# TD-JEPA: Latent-predictive Representations for Zero-Shot Reinforcement Learning

**Marco Bagatella**[123*]**, Matteo Pirotta**[1]**, Ahmed Touati**[1]**, Alessandro Lazaric**[1]**, Andrea Tirinzoni**[1]
[1] FAIR at Meta, [2] ETH Zürich, [3] Max Planck Institute for Intelligent Systems, Tübingen
`mbagatella@ethz.ch`, {`pirotta,touati,lazaric,tirinzoni`}`@meta.com`

## Abstract

Latent prediction–where agents learn by predicting their own latents–has emerged as a powerful paradigm for training general representations in machine learning. In reinforcement learning (RL), this approach has been explored to define auxiliary losses for a variety of settings, including reward-based and unsupervised RL, behavior cloning, and world modeling. While existing methods are typically limited to single-task learning, one-step prediction, or on-policy trajectory data, we show that temporal difference (TD) learning enables learning representations predictive of long-term latent dynamics across multiple policies/tasks from offline, reward-free transitions. Building on this, we introduce TD-JEPA, which leverages TD-based latent-predictive representations into unsupervised RL. TD-JEPA trains explicit state and task encoders, a policy-conditioned multi-step predictor, and a set of parameterized policies directly in latent space. This enables zero-shot optimization of any reward function at test time. Theoretically, we show that an idealized variant of TD-JEPA avoids collapse with proper initialization, and learns encoders that capture a low-rank factorization of long-term policy dynamics, while the predictor recovers their successor features in latent space. Empirically, TD-JEPA matches or outperforms state-of-the-art baselines on locomotion, navigation, and manipulation tasks across 13 datasets in ExoRL and OGBench, especially in the challenging setting of zero-shot RL from pixels.[1]

## 1 Introduction

Learning effective state representations is a core challenge in reinforcement learning (RL). Useful representations should capture the dynamics of the environment in a way that supports efficient value estimation and policy optimization across tasks (Watter et al., 2015; Silver et al., 2018; Hafner et al., 2019; Gelada et al., 2019). A promising line of work is latent-predictive (a.k.a. self-predictive) representation learning (Schwarzer et al., 2021; Grill et al., 2020; Guo et al., 2020; Tang et al., 2023), an instance of the joint-embedding predictive architecture (LeCun, 2022, JEPA) paradigm. These algorithms jointly learn a *state encoder* $\phi(s)$ and a *predictor* $P$, i.e., a latent dynamics model estimating the representation of a future state $s'$: $P(\phi(s)) \simeq \phi(s')$. Latent-predictive methods thus perform *self-supervised* learning entirely in latent space without any reward or reconstruction of (possibly high-dimensional) states.

Several RL methods leverage latent prediction as an auxiliary loss to improve sample efficiency and generalization in reward-based learning (Schwarzer et al., 2021; Guo et al., 2020; Hansen et al., 2024), behavior cloning (Lawson et al., 2025), and curiosity-driven exploration (Guo et al., 2022). As latent-predictive losses do not require any reward, they have been recently used for unsupervised RL: Assran et al. (2025a), Zhou et al. (2025) and Sobal et al. (2025) learn latent world models that can solve goal-reaching tasks via test-time planning, whereas Jajoo et al. (2025) learn a state encoder from trajectory data to define the space of tasks used to optimize zero-shot unsupervised policies.

This paper proposes a novel way to instantiate latent-predictive representations for unsupervised RL. While previous methods have largely focused on either one-step dynamics, single-task/single-

---

*Work done at Meta.
[1]Code available at github.com/facebookresearch/td_jepa.

Figure 1: TD-JEPA trains policies $\pi_z$ parameterized by latents $z$. The predictor, conditioned on $z$, predicts the representations of future states visited by $\pi_z$ (*left*). When trained via TD, the predictor (arrows on the *right*) approximates successor features for each policy, i.e., the weighted barycenter (stars) of representations of visited states (circles).

policy training, or relied on on-policy data, we introduce a policy-conditioned, multi-step formulation based on a novel off-policy temporal-difference loss. This objective encourages representations that are predictive not only of immediate transitions, but also of long-term features relevant for value estimation across multiple policies. This property makes such representations and the associated predictors particularly well-suited for integration with off-policy, successor-feature based approaches to zero-shot unsupervised RL (Touati & Ollivier, 2021; Touati et al., 2023; Park et al., 2024), which have recently emerged as a promising solution for applications such as whole-body humanoid control in simulation (Tirinzoni et al., 2025) and on real hardware (Li et al., 2025).

We thus instantiate temporal difference latent-predictive representation learning into TD-JEPA, a zero-shot unsupervised RL algorithm which pre-trains four components: a state encoder, a policy-conditioned multi-step predictor, a task encoder, and a set of parameterized policies, all of which are learned end-to-end from offline, reward-free transitions. Departing from previous approaches, latent prediction is not merely an auxiliary loss, but rather the core objective that enables TD-JEPA to learn all the components needed to distill zero-shot policies. In fact, the predictor may be leveraged as an approximation of successor features (see Figure 1) to extract policies mapping encoded observations to optimal actions for all reward functions in the span of the learned features. This enables TD-JEPA to perform zero-shot policy optimization for any downstream reward, *entirely in latent space*.

Theoretically, for an idealized version of TD-JEPA with linear predictors, we show that **1)** the representations do not collapse with a suitable initialization; **2)** they recover a low-rank factorization of the successor measures of the trained policies, while the predictor approximates successor features in latent space; **3)** they minimize an upper bound on the policy evaluation error for any reward, thus making zero-shot optimization possible. These results build on a novel "gradient matching" argument that extends and generalizes existing theoretical analyses of latent-predictive representations, and connect TD-JEPA with other unsupervised RL methods such as forward-backward (Touati & Ollivier, 2021) and intention-conditioned value functions (Ghosh et al., 2023).

Empirically, we evaluate TD-JEPA on 65 tasks across 13 datasets from ExoRL (Yarats et al., 2022a) and OGBench (Park et al., 2025a), covering locomotion, navigation, and manipulation with both proprioceptive and pixel-based observations. TD-JEPA matches or outperforms state-of-the-art zero-shot baselines across these settings, in particular when learning from pixels, which has proven to be one of the most challenging settings for unsupervised RL so far. Moreover, we ablate several dimensions of the algorithm, demonstrating the importance of learning representations that are predictive of multi-step policy-dependent dynamics, and the advantage of training distinct state and task encoders. Finally, we show that learned representations can be easily reused for offline or online RL, improving over zero-shot policies and learning from scratch.

## 2 PRELIMINARIES

We consider a reward-free Markov Decision Process $\mathcal{M} = (\mathcal{S}, \mathcal{A}, P, \gamma)$, where $\mathcal{S}$ and $\mathcal{A}$ are state and action spaces, $P$ is the probability measure over next states when taking action $a$ in state $s$ as $P(\mathrm{d}s' \mid s, a)$, and $\gamma \in [0, 1)$ is a discount factor. Executing a Markov policy $\pi : \mathcal{S} \to \mathrm{Prob}(\mathcal{A})$ induces an unnormalized distribution over visited states, which is referred to as the *successor measure*:

$$M^\pi(\mathcal{X} \mid s, a) = \sum_{t=0}^{\infty} \gamma^t \mathrm{Pr}(s_{t+1} \in \mathcal{X} \mid s, a, \pi) \quad \forall \, \mathcal{X} \subseteq \mathcal{S}. \tag{1}$$

Given a reward function $r : \mathcal{S} \rightarrow \mathbb{R}$ and a policy $\pi$, the action-value function $Q_r^\pi(s, a)$ measures the cumulative discounted reward obtained by the policy over an infinite horizon, i.e., $Q_r^\pi(s, a) = \mathbb{E}\left[ \sum_{t=0}^\infty \gamma^t r(s_{t+1}) \mid s, a, \pi \right]$. Action-value functions are connected to successor measures via

$$Q_r^\pi(s, a) = \int_{s^+ \in \mathcal{S}} M^\pi(\mathrm{d}s^+ \mid s, a) r(s^+) = \mathbb{E}_{s^+ \sim M^\pi(\cdot \mid s, a)}\left[ r(s^+) \right], \tag{2}$$

which shows a convenient linear decomposition of $Q_r^\pi$ into the reward function and the dynamics induced by $\pi$. Standard RL agents aim at finding reward-maximizing policies $\pi_r^\star(s) \in \arg\max_{a \in \mathcal{A}} Q_r^\star(s, a)$, where $Q_r^\star(s, a) := \max_\pi Q_r^\pi(s, a)$.

**Latent-predictive representations.** In high-dimensional settings, *state encoders* $\phi : \mathcal{S} \rightarrow \mathbb{R}^{d_\phi}$ may be learned to ease the estimation of action-value functions. For instance, if an encoder $\phi$ is such that $Q_r^\pi(s, a) = \phi(s)^\top w_{a,r}^\pi$ for some vector $w_{a,r}^\pi \in \mathbb{R}^{d_\phi}$, then the RL process reduces to learning vectors in $\mathbb{R}^{d_\phi}$ rather than high-dimensional functions $Q_r^\pi(s, a)$. Latent-predictive learning has been shown to be an effective approach for this problem. In the simplest formulation, latent-predictive representations capture the one-step latent dynamics of a policy $\pi$ by minimizing the loss

$$\mathcal{L}_{\text{one-step}}(\phi, T) = \mathbb{E}_{s \sim \rho, a \sim \pi(\cdot \mid s), s' \sim P(\cdot \mid s, a)}\left[ \| T(\phi(s)) - \overline{\phi(s')} \|^2 \right], \tag{3}$$

where $T : \mathbb{R}^{d_\phi} \rightarrow \mathbb{R}^{d_\phi}$ is a (possibly non-linear) predictor of the latent one-step dynamics induced by $\phi$ and policy $\pi$, and $\overline{\phi}$ denotes stop-gradient. Notably, optimizing for this loss does not require any decoding or reconstruction, and it only relies on an *unsupervised* dataset $\mathcal{D} = \{(s, a, s')\}$. Different instantiations of this approach have been shown both empirically and theoretically to produce representations that accurately approximate action-value functions or policies (Guo et al., 2022; Tang et al., 2023; Voelcker et al., 2024; Lawson et al., 2025; Fujimoto et al., 2025).

**Successor-features and zero-shot unsupervised RL.** Considering a state encoder $\psi : \mathcal{S} \rightarrow \mathbb{R}^{d_\psi}$ and the associated space of linear rewards $\mathcal{R}_\psi = \{r(s) = \psi(s)^\top z \mid z \in \mathbb{R}^{d_\psi}\}$, Q-values for any reward function $r(s) = \psi(s)^\top z_r \in \mathcal{R}_\psi$ can be written as

$$Q_r^\pi(s, a) = \int_{s^+ \in \mathcal{S}} M^\pi(\mathrm{d}s^+ \mid s, a) \psi(s^+)^\top z_r = \mathbb{E}_{s^+ \sim M^\pi(\cdot \mid s, a)}\left[ \psi(s^+) \right]^\top z_r := F_\psi^\pi(s, a)^\top z_r, \tag{4}$$

where $F_\psi^\pi(s, a) \in \mathbb{R}^{d_\psi}$ captures the *successor features* of $\pi$ (Barreto et al., 2017). The majority of unsupervised zero-shot RL methods (Touati & Ollivier, 2021; Park et al., 2024; Agarwal et al., 2025; Jajoo et al., 2025) learn successor features $F(s, a; z) \approx F_\psi^{\pi_z}(s, a)$ for a set of parameterized policies $\{\pi_z(s)\}_{z \in \mathcal{Z}}$, with $\mathcal{Z} \subseteq \mathbb{R}^d$, that are trained to be optimal for all rewards in $\mathcal{R}_\psi$, i.e., $\pi_z(s) \approx \arg\max_a F(s, a; z)^\top z$, where $F(s, a; z)^\top z$ is an approximation of $Q_r^\star(s, a)$ for $r(s) = \psi(s)^\top z$. At test time, given a reward function $r$, a vector $z_r \in \mathbb{R}^{d_\psi}$ is first obtained by projecting $r$ onto $\mathcal{R}_\psi$, and the associated policy $\pi_{z_r}$ is then returned.

Given the role played by $\psi$ in defining the space of tasks of interest, with an abuse of terminology, we will refer to $\psi$ as a *task encoder*. On the other hand, we shall call *state encoder* a map $\phi : \mathcal{S} \rightarrow \mathbb{R}^{d_\phi}$ that is used to embed states before feeding them into different networks (e.g., we will train successor features $F_\psi^\pi(\phi(s), a)$ and policies $\pi(\phi(s))$ in the latent space given by $\phi$). While the zero-shot methods cited so far train the task encoder $\psi$ in different ways, and do not train any explicit state encoder $\phi$, the next section will show how multi-step policy-dependent latent-predictive learning can be used to train both simultaneously.

## 3 LATENT-PREDICTIVE TEMPORAL-DIFFERENCE REPRESENTATIONS

We begin by showing how the latent-predictive loss of Eq. 3 can model multi-step and policy-dependent dynamics, and how temporal difference (TD) learning allows learning from offline transition data. We will then expand this idea to learn separate state and task embeddings, and finally show how it can be instantiated as a zero-shot unsupervised RL method.

### 3.1 MULTI-STEP POLICY-CONDITIONED LATENT PREDICTION

Let $\{\pi_z\}_{z \in \mathcal{Z}}$ be a family of policies parameterized by $z \in \mathcal{Z}$, and $\mathcal{D} = \{(s, a, s')\}$ be a dataset of transitions. We train a state encoder $\phi : \mathcal{S} \rightarrow \mathbb{R}^{d_\phi}$ and a *policy-dependent* predictor $T_\phi :$

$\mathbb{R}^{d_\phi} \times \mathcal{A} \times \mathcal{Z} \to \mathbb{R}^{d_\phi}$ to be latent-predictive of the *long-term* dynamics of the policies $\{\pi_z\}$, i.e.,

$$\mathcal{L}_{\text{MC-JEPA}}(\phi, T_\phi) = \mathbb{E}_{(s,a)\sim\mathcal{D}, z\sim\mathcal{Z}, s^+\sim M^{\pi_z}(\cdot|s,a)}\big[\|T_\phi(\phi(s), a, z) - \overline{\phi(s^+)}\|^2\big], \qquad (5)$$

where *MC-JEPA* stands for Monte-Carlo (MC) JEPA loss, as on-policy samples $s^+ \sim M^{\pi_z}(\cdot|s,a)$ are needed for all policies of interest. Intuitively, $T_\phi(\phi(s), a, z)$ tries to predict future latent states visited by the policy $\pi_z$. More formally, predictors trained via minimization of $\mathcal{L}_{\text{MC-JEPA}}(\phi, T_\phi)$ approximate the successor features of $\phi$ in the latent space induced by $\phi$ itself.

**Proposition 1.** *For any $\phi$ and $T_\phi$, we have the following equivalence*

$$\mathcal{L}_{\text{MC-JEPA}}(\phi, T_\phi) = \mathbb{E}_{(s,a)\sim\mathcal{D}, z\sim\mathcal{Z}}\big[\|T_\phi(\phi(s), a, z) - \overline{F_\phi^{\pi_z}(s,a)}\|^2\big] + \text{const.} \qquad (6)$$

Given the connection between Q-functions and successor features (Eq. 4), this result crucially relates multi-step latent prediction with value estimation across multiple policies. More precisely, it implies that the predictor enables policy evaluation and optimization of rewards in the span of $\phi$, as we detail at the end of this section. Since $F_\phi^{\pi_z}$ is the successor features of $\phi$, with the terminology introduced in Sec. 2, $\phi$ is used both as a *state encoder*, i.e., to embed states passed to the predictor, and as a *task encoder*, i.e., defining a space of reward functions.

Unfortunately, this loss cannot be estimated on off-policy data since it requires sampling from the successor measures of the given policies. We can however leverage the previous result and the fact that successor features admit a Bellman equation $F_\phi^{\pi_z}(s, a) = \mathbb{E}_{s'\sim P(\cdot|s,a), a'\sim\pi_z(s')}[\phi(s') + \gamma F_\phi^{\pi_z}(s', a')]$ (Barreto et al., 2017) to define a temporal-difference version of the previous loss:

$$\mathcal{L}_{\text{TD-JEPA}}(\phi, T_\phi) = \mathbb{E}_{(s,a,s')\sim\mathcal{D}, z\sim\mathcal{Z}, a'\sim\pi_z(\cdot|s')}\big[\|T_\phi(\phi(s), a, z) - \overline{\phi(s')} - \gamma\overline{T_\phi(\phi(s'), a', z)}\|^2\big]. \quad (7)$$

Unlike the Monte Carlo loss of Eq. 5, $\mathcal{L}_{\text{TD-JEPA}}$ only requires sampling one-step transitions and actions from the given policies, and it can thus be estimated from off-policy, offline datasets.

## 3.2 TRAINING SEPARATE STATE AND TASK REPRESENTATIONS

While in Eq. 5 and 7 the same encoder $\phi$ is used for both state and task representations, these need not be the same in practice. Consider, for instance, a robot navigating a building: useful state representations may capture low-level dynamical information critical for control (e.g., joint positions and velocities), while task representations could abstract higher-level contextual features, such as the building's topology. In this case, a single representation might be either too complex, or too abstract: having flexibility over the dimensionality and content of each representation would be desirable. We thus now introduce an asymmetric variant that trains a distinct encoder $\psi : \mathcal{S} \to \mathbb{R}^{d_\psi}$ to define the set of reward functions of interest (i.e., as a *task* encoder). We first redefine the predictor as $T_\phi : \mathbb{R}^{d_\phi} \times \mathcal{A} \times \mathcal{Z} \to \mathbb{R}^{d_\psi}$ and the latent-predictive Monte-Carlo loss to train $\phi$ and $T_\phi$ as

$$\mathcal{L}_{\text{MC-JEPA}}(\phi, T_\phi, \psi) = \mathbb{E}_{(s,a)\sim\mathcal{D}, z\sim\mathcal{Z}, s^+\sim M^{\pi_z}(\cdot|s,a)}\big[\|T_\phi(\phi(s), a, z) - \overline{\psi(s^+)}\|^2\big], \qquad (8)$$

such that $T_\phi$ maps states encoded through $\phi$ to the long-term dynamics of a policy $\pi_z$ in the latent space induced, this time, by $\psi$. Similar to Prop. 1, $T_\phi$ approximates the successor features $F_\psi^{\pi_z}(s, a)$ of $\psi$ in the latent space induced by $\phi$. Symmetrically, we train $\psi$ together with an additional predictor $T_\psi : \mathbb{R}^{d_\psi} \times \mathcal{A} \times \mathcal{Z} \to \mathbb{R}^{d_\phi}$. To do so, we follow existing literature – according to which joint representations should be predictive of each other (Guo et al., 2020; Tang et al., 2023) – and train $\psi$ and $T_\psi$ through the same latent-predictive loss with the roles of $\phi$ and $\psi$ inverted, i.e., $\mathcal{L}_{\text{MC-JEPA}}(\psi, T_\psi, \phi)$.[2] As before, we can then design an off-policy TD variant of this loss,

$$\mathcal{L}_{\text{TD-JEPA}}(\phi, T_\phi, \psi) = \mathbb{E}_{\substack{(s,a,s')\sim\mathcal{D} \\ z\sim\mathcal{Z}, a'\sim\pi_z(\cdot|s')}} \big[\|T_\phi(\phi(s), a, z) - \overline{\psi(s')} - \gamma\overline{T_\phi(\phi(s'), a', z)}\|^2\big], \qquad (9)$$

so that $\phi$ and $T_\phi$ are optimized via $\mathcal{L}_{\text{TD-JEPA}}(\phi, T_\phi, \psi)$, while $\psi$ and $T_\psi$ via $\mathcal{L}_{\text{TD-JEPA}}(\psi, T_\psi, \phi)$.

## 3.3 TD-JEPA REPRESENTATIONS FOR ZERO-SHOT RL

The relationship between the learned predictors and successor features suggests a seamless instantiation of TD-JEPA as a zero-shot unsupervised RL algorithm. Redefining the policy parameter

---

[2]While some existing works use forward-in-time sampling to train one representation and backward-in-time for the other, we use two forward-in-time losses. We further discuss this difference in App. C.

---

**Algorithm 1** TD-JEPA for zero-shot RL

---

**Inputs**: Dataset $\mathcal{D}$, batch size $B$, regularization coefficient $\lambda$, networks $\pi, T_\phi, \phi, T_\psi, \psi$
Initialize target networks: $T_\phi^- \leftarrow T_\phi, \phi^- \leftarrow \phi, T_\psi^- \leftarrow T_\psi, \psi^- \leftarrow \psi$
**while** not converged **do**

$\quad\triangleright$ Sample training batch
$\quad \{(s_i, a_i, s_i')\}_{i=1}^B \sim \mathcal{D}, \{z_i\}_{i=1}^B \sim \mathcal{Z}, \{a_i'\}_{i=1}^B \sim \overline{\{\pi(\phi^-(s_i'), z_i)\}}_{i=1}^B$

$\quad\triangleright$ Compute latent-predictive losses
$\quad \widehat{\mathcal{L}}_{\text{TD-JEPA}}(\phi, T_\phi, \psi) = \frac{1}{2B}\sum_i \left\| T_\phi(\phi(s_i), a_i, z_i) - \overline{\psi^-(s_i')} - \gamma \overline{T_\phi^-(\phi^-(s_i'), a_i', z_i)} \right\|^2$
$\quad \widehat{\mathcal{L}}_{\text{TD-JEPA}}(\psi, T_\psi, \phi) = \frac{1}{2B}\sum_i \left\| T_\psi(\psi(s_i), a_i, z_i) - \overline{\phi^-(s_i')} - \gamma \overline{T_\psi^-(\psi^-(s_i'), a_i', z_i)} \right\|^2$

$\quad\triangleright$ Compute orthonormality regularization losses
$\quad \widehat{\mathcal{L}}_{\text{REG}}(\phi) = \frac{1}{2B(B-1)}\sum_{i \neq j}(\phi(s_i)^\top \phi(s_j))^2 - \frac{1}{B}\sum_i \phi(s_i)^\top \phi(s_i)$
$\quad \widehat{\mathcal{L}}_{\text{REG}}(\psi) = \frac{1}{2B(B-1)}\sum_{i \neq j}(\psi(s_i)^\top \psi(s_j))^2 - \frac{1}{B}\sum_i \psi(s_i)^\top \psi(s_i)$

$\quad\triangleright$ Compute actor loss
$\quad \{\hat{a}_i\}_{i=1}^B \sim \{\pi(\phi(s_i), z_i)\}_{i=1}^B$
$\quad \widehat{\mathcal{L}}_{\text{actor}}(\pi) = -\frac{1}{B}\sum_{i=1}^B T_\phi(\phi(s_i), \hat{a}_i, z_i)^\top z_i$

$\quad$ Update $\phi, T_\phi$ to minimize $\widehat{\mathcal{L}}_{\text{TD-JEPA}}(\phi, T_\phi, \psi) + \lambda \widehat{\mathcal{L}}_{\text{REG}}(\phi)$
$\quad$ Update $\psi, T_\psi$ to minimize $\widehat{\mathcal{L}}_{\text{TD-JEPA}}(\psi, T_\psi, \phi) + \lambda \widehat{\mathcal{L}}_{\text{REG}}(\psi)$
$\quad$ Update $\pi$ to minimize $\widehat{\mathcal{L}}_{\text{actor}}(\pi)$
$\quad$ Update target networks $\phi^-, T_\phi^-, \psi^-, T_\psi^-$ via EMA of $\phi, T_\phi, \psi, T_\psi$

---

space $\mathcal{Z}$ as the task embedding space (i.e., $\mathcal{Z} \subseteq \mathbb{R}^{d_\psi}$), we train latent policies such that $\pi_z(\phi(s)) = \arg\max_a T_\phi(\phi(s), z, a)^\top z$ for all $z \in \mathcal{Z}^3$. Since $T_\phi(\phi(s), z, a) \simeq F_\psi^{\pi_z}(s, a)$ (Proposition 1), this produces optimal policies for all rewards in the span of $\psi$, learned directly from state representations $\phi(\cdot)$. At test time, given an inference dataset of rewarded samples $\mathcal{D}_{\text{rwd}} = \{(s, r)\}$, the optimal policy $\pi_{z_r}$ can be retrieved by computing $z_r$ through linear regression, e.g. through the closed-form solution $z_r = \arg\min_z \mathbb{E}_{(s,r)\sim\mathcal{D}_{\text{rwd}}}[(r - \psi(s)^\top z)^2] = \mathbb{E}_{s\sim\mathcal{D}_{\text{rwd}}}[\psi(s)\psi(s)^\top]^{-1}\mathbb{E}_{(s,r)\sim\mathcal{D}_{\text{rwd}}}[\psi(s)r(s)]$. Alg. 1 describes TD-JEPA, which combines $\mathcal{L}_{\text{TD-JEPA}}$ with stabilization strategies, e.g. target networks and covariance regularization. We remark that latent prediction is not auxiliary: it is the core objective that trains encoders and predictors, from which zero-shot policies can be directly distilled.

## 4 THEORETICAL ANALYSIS

We now provide some theoretical arguments showing how latent-predictive temporal difference representations capture the long-term dynamics of a given set of policies in a way that makes them amenable to zero-shot RL. Following Tang et al. (2023), we consider a simplified tabular setting with linear predictors. We view the representation $\phi$ (resp. $\psi$) as a $S \times d_\phi$ (resp. $S \times d_\psi$) matrix, and consider action-free predictors $T_{\phi,z}$ (resp. $T_{\psi,z}$) as $d_\phi \times d_\psi$ (resp. $d_\psi \times d_\phi$) matrices for all $z$. The expression $T_\phi(\phi(s), a, z)$ in Eq. 8 and 9 thus reduces to $T_{\phi,z}^\top \phi(s)$, while $M^\pi(s'|s, a)$ and $P(s'|s, a)$ are replaced by $M^{\pi_z}(s'|s) = M^{\pi_z}(s'|s, \pi_z(s))$ and $P^{\pi_z}(s'|s) = P(s'|s, \pi_z(s))$.

**Monte-Carlo losses.** We define a (non-latent-predictive) successor measure approximation loss

$$\mathcal{L}_{\text{SM}}(\phi, \{T_z\}_z, \psi) := \frac{1}{2}\mathbb{E}_{z\sim\mathcal{Z}}\|\phi T_z \psi^\top - M^{\pi_z}\|_F^2. \tag{10}$$

Minimizing $\mathcal{L}_{\text{SM}}$ is equivalent to finding the best multilinear approximation to the successor measures $M^{\pi_z}$. We prove the following connection with the Monte Carlo latent-predictive loss of Eq. 8.

**Theorem 1.** *For fixed $\phi$ and $\psi$, let $T_z^\star, T_{\phi,z}^\star, T_{\psi,z}^\star$ be the optimal predictors for $\mathcal{L}_{\text{SM}}(\phi, T_z, \psi)$ (Eq. 10), $\mathcal{L}_{\text{MC-JEPA}}(\phi, T_{\phi,z}, \psi), \mathcal{L}_{\text{MC-JEPA}}(\psi, T_{\psi,z}, \phi)$ (Eq. 8), respectively. If (A1) $\phi^\top\phi = \psi^\top\psi = I$, (A2) the state distribution is uniform, and (A3) for all $z \in \mathcal{Z}$, the matrix $P^{\pi_z}$ is symmetric, then*

---

[3]This decision additionally grounds $\psi$ as task encoder, and breaks the symmetry that could arise from the two encoders $\phi$ and $\psi$ being trained through similar latent-predictive objectives.

1. *for all $z$, $\phi T_z^\star = \phi T_{\phi,z}^\star = \Pi_\phi M^{\pi_z} \psi$ and $\psi T_{\psi,z}^\star = \psi(T_z^\star)^\mathsf{T} = \Pi_\psi M^{\pi_z} \phi$, where $\Pi_\phi$ (resp. $\Pi_\psi$) is an orthogonal projection on the span of $\phi$ (resp. $\psi$);*
2. $\nabla_\phi \mathcal{L}_{\text{MC-JEPA}}(\phi, T_z, \psi) = \nabla_\phi \mathcal{L}_{\text{SM}}(\phi, T_z, \psi)$ *and* $\nabla_\psi \mathcal{L}_{\text{MC-JEPA}}(\psi, T_z, \phi) = \nabla_\psi \mathcal{L}_{\text{SM}}(\phi, T_z^\mathsf{T}, \psi)$.

This result reveals that **1)** the optimal predictors for the successor measure loss $\mathcal{L}_{\text{SM}}$ and the latent-predictive loss $\mathcal{L}_{\text{MC-JEPA}}$ match, and yield an orthogonal projection of the successor features $M^{\pi_z}\psi$ onto the $\phi$ space; **2)** the gradients w.r.t. the representations $\phi$ and $\psi$, when evaluated at any predictor, match among these two losses, showing that gradient descent on $\mathcal{L}_{\text{MC-JEPA}}$ would update representations in the direction that reduces $\mathcal{L}_{\text{SM}}$, hence improving the approximation of the successor measures. This result follows as a special case of a novel theorem (see App. C) generalizing and implying all previous guarantees for latent-predictive representations (Tang et al., 2023; Khetarpal et al., 2025; Voelcker et al., 2024; Lawson et al., 2025), which we believe is of independent interest. Finally, we remark that, while the assumptions A1-A3 have been considered in all these related works, they can be relaxed, at the price of more involved proofs and notation, as shown in App. C.

**Temporal-difference losses.** We first derive a non-collapse guarantee. While a similar result was originally proved by Tang et al. (2023) for the one-step loss (Eq. 3), our case is more complex since TD latent-prediction can be seen as "doubly latent-predictive" (cf. Eq. 9): $T_{\phi,z}^\mathsf{T}\phi(s)$ is optimized to match a representation being learned – $\psi(s^+)$ – plus a bootstrapped version of itself – $T_{\phi,z}^\mathsf{T}\phi(s^+)$.

**Theorem 2.** *Let $\phi_t$ and $\psi_t$ be the representations learned under a continuous-time relaxation of Eq. 9 where, at each step $t$, the optimal predictors for $(\phi_t, \psi_t)$ are first computed and then a gradient step on $(\phi_t, \psi_t)$ is taken (see App. B.3 for the explicit formulation). Then, the covariance matrices $\phi_t^\mathsf{T}\phi_t$ and $\psi_t^\mathsf{T}\psi_t$ are constant over time, i.e., $\phi_t^\mathsf{T}\phi_t = \phi_0^\mathsf{T}\phi_0$ and $\psi_t^\mathsf{T}\psi_t = \psi_0^\mathsf{T}\psi_0$ for all $t \geq 0$.*

This result suggests that, if predictors are trained at a faster rate than representations, the overall dynamics preserve their covariance, thus preventing $\phi$ and $\psi$ from collapsing to trivial solutions (e.g., $\phi = \psi = 0$) when properly initialized, e.g., with unitary covariance.

As done for MC objectives (Th. 1), we now show that the latent-predictive loss of TD-JEPA is related to forward and backward TD losses for approximating the successor measure (Blier et al., 2021).

**Theorem 3.** *Consider the following TD losses for approximating the successor measure*

$$\mathcal{L}_{\text{fw}}(\phi, T_z, \psi) := \frac{1}{2}\mathbb{E}_{z \sim \mathcal{Z}}\left[\|\phi T_z \psi^\mathsf{T} - P^{\pi_z} - \gamma\overline{P^{\pi_z}\phi T_z \psi^\mathsf{T}}\|_F^2\right], \tag{11}$$

$$\mathcal{L}_{\text{bw}}(\phi, T_z, \psi) := \frac{1}{2}\mathbb{E}_{z \sim \mathcal{Z}}\left[\|\psi T_z \phi^\mathsf{T} - (P^{\pi_z})^\mathsf{T} - \gamma(P^{\pi_z})^\mathsf{T}\overline{\psi T_z \phi^\mathsf{T}}\|_F^2\right]. \tag{12}$$

*For fixed $(\phi, \psi)$, let $T_{z,\text{fw}}^\star, T_{z,\text{bw}}^\star, T_{\phi,z}^\star, T_{\psi,z}^\star$ respectively be the optimal predictors for $\mathcal{L}_{\text{fw}}(\phi, T_z, \psi)$, $\mathcal{L}_{\text{bw}}(\phi, T_z, \psi)$, $\mathcal{L}_{\text{TD-JEPA}}(\phi, T_z, \psi)$, $\mathcal{L}_{\text{TD-JEPA}}(\psi, T_z, \phi)$. Under the same assumptions as Th. 1,*

1. *for all $z$, $\phi T_{\phi,z}^\star = \phi T_{z,\text{fw}}^\star = \tilde{\Pi}_{\phi,z} M^{\pi_z} \psi$ and $\psi T_{\psi,z}^\star = \psi T_{z,\text{bw}}^\star = \tilde{\Pi}_{\psi,z} M^{\pi_z} \phi$, where $\tilde{\Pi}_{\phi,z}$ (resp. $\tilde{\Pi}_{\psi,z}$) is an oblique projection on the span of $\phi$ (resp. $\psi$);*
2. $\nabla_\phi \mathcal{L}_{\text{TD-JEPA}}(\phi, T_z, \psi) = \nabla_\phi \mathcal{L}_{\text{fw}}(\phi, T_z, \psi)$ *and* $\nabla_\psi \mathcal{L}_{\text{TD-JEPA}}(\psi, T_z, \phi) = \nabla_\psi \mathcal{L}_{\text{fw}}(\phi, T_z, \psi)$.

Similar to Th. 1, the optimal predictors and gradients of TD-JEPA match those of the non-latent-predictive TD losses of Eq. 11 and 12, which are known to recover an approximation of the successor measure for bilinear parameterizations of the form $F_z^\mathsf{T}B$ (Blier et al., 2021). Unlike in the Monte Carlo case, here the optimal predictors solve a least-squares TD problem (Boyan, 1999; Precup et al., 2001), yielding the fixed point of a projected Bellman operator whose closed-form expression is an oblique projection (Scherrer, 2010).

**Policy evaluation and zero-shot RL.** Finally, the following result motivates the significance of optimizing the successor measure losses of Eq. 10, 11, and 12.

**Theorem 4.** *Let $\phi, \psi$ have identity covariance matrices. For any reward function $r$, let $\omega_r := (\psi^\mathsf{T}\psi)^{-1}\psi^\mathsf{T}r$ be the linear regression weight for representation $\psi$. Then, for any $T_z$,*

$$\max_{r \in \mathbb{R}^S : \|r\|_2 \leq 1} \mathbb{E}_{z \in \mathcal{Z}}\left[\sum_{s \in \mathcal{S}}\left(V_r^{\pi_z}(s) - \phi(s)^\mathsf{T}T_z\omega_r\right)^2\right] \leq 2\mathcal{L}_{\text{SM}}(\phi, T_z, \psi).$$

*Moreover, $\mathcal{L}_{\text{SM}}(\phi, T_z, \psi) \leq c\mathcal{L}_{\text{fw}}(\phi, T_z, \psi)$ and $\mathcal{L}_{\text{SM}}(\phi, T_z, \psi) \leq c\mathcal{L}_{\text{bw}}(\phi, T_z, \psi)$ for some $c$.*

Paraphrasing, the policy evaluation error of the technique in Section 3.3 (i.e., embed $r$ into a vector $\omega$ through linear regression on $\psi$, and compute $T_\phi(\phi(s), z)^\mathsf{T} \omega)$ is bounded by the successor measure approximation loss and the corresponding TD errors. Both these quantities are indirectly optimized by TD-JEPA (Th. 1, 3), which is thus a sound approach for zero-shot policy evaluation. Moreover, Th. 4 leads to a zero-shot optimality result analogous to Theorem 2 of (Touati & Ollivier, 2021): if the approximation of $M^{\pi_z}$ is perfect (i.e., $M^{\pi_z} = \phi T_z \psi^\mathsf{T}$ for all $z$ or, equivalently, the TD errors in Eq. 11 and 12 are zero) and the policies $\pi_z$ are optimal for all linear rewards in $\psi$, then the inference procedure above recovers optimal policies for *any* (even non-linear) reward function.

## 5 RELATED WORK

**Zero-shot RL algorithms.** Methods that pre-train agents on unsupervised data to enable zero-shot solution of a wide range of downstream tasks have achieved impressive results, yielding so-called behavioral foundation models (Pirotta et al., 2024; Tirinzoni et al., 2025). The forward-backward algorithm (FB, Touati & Ollivier (2021); Touati et al. (2023)) is an established method, and perhaps the most related to TD-JEPA. FB learns a task encoder and estimates its successor features, essentially finding a bilinear decomposition of policy-conditional successor measures (e.g., $M^{\pi_z} \approx F_z B^\mathsf{T}$). On the other hand, TD-JEPA uses the parameterization $M^{\pi_z} \approx \phi T_z \psi^\mathsf{T}$, thus training shared (across tasks) state representations. Moreover, FB adopts a *contrastive* loss, which computes pairwise dot products across each training batch, while TD-JEPA is non-contrastive at its core. FB has been further shown capable of zero-shot imitation (Pirotta et al., 2024) and extended to several settings, including online training regularized by action-free expert data (Tirinzoni et al., 2025), offline training on low-quality data (Jeen et al., 2024b), training on environments with different dynamics (Bobrin et al., 2025), online fine-tuning (Sikchi et al., 2025) and pure exploration (Urpí et al., 2025). Other methods, like HILP, PSM, and RLDP, can also be seen as training a task encoder $\psi$ plus successor features on top. HILP (Park et al., 2024) trains $\psi$ through a distance-preserving "goal-reaching" loss, while PSM (Agarwal et al., 2025) learns an affine decomposition of the successor measure for a discrete codebook of policies, and RLDP (Jajoo et al., 2025) trains $\psi$ using chained multi-step latent prediction (Hansen et al., 2024). Jajoo et al. (2025) also observe that regularizing the representation to be orthonormal is crucial to avoid collapse, which we also observe in TD-JEPA (see Alg. 1).

**Latent-predictive methods.** Latent-predictive methods have mostly been applied to define auxiliary losses for a variety of RL settings. Schwarzer et al. (2021) use a latent-predictive loss to enhance state representations learned through a deep Q network. Guo et al. (2020) use latent prediction in POMDPs to encourage two representations (of observations and histories) to be self-predictive of each other, similarly to the asymmetric variant of TD-JEPA and the method explained in Appendix C. Hansen et al. (2024) and Sobal et al. (2025) train a latent dynamics model that enables test-time planning to improve a pre-trained policy or solve goal-reaching tasks, respectively. BYOL-$\gamma$ (Lawson et al., 2025) trains representations by predicting discounted future latent states visited by the behavior policy. BYOL-$\gamma$ may thus be seen as an unconditional, Monte Carlo version of TD-JEPA, which is instead policy-conditional and off-policy. The on-policy nature of the algorithm enables Lawson et al. (2025) to implement a bi-directional update of asymmetric representations. TD-JEPA can also recover an asymmetric parameterization, but its practical objective is not bi-directional (i.e., it only implements forward TD prediction, cf. Appendix C.3 for a formal definition of the bi-directional objective). Crucially, BYOL-$\gamma$ is not proposed as a zero-shot method: the version we evaluate is a novel instantiation in a successor-feature framework.

**Theory of latent-predictive representations.** The theory of latent-predictive representations has been previously studied in several works (Tang et al., 2023; Voelcker et al., 2024; Khetarpal et al., 2025; Lawson et al., 2025), with a particular focus on single-policy, single-step prediction (potentially, bi-directional). Our analysis of MC-JEPA (Section 4) largely takes place in a multi-policy setting, with generic transition kernels over states; as such, it subsumes and expands on several existing results (see Appendix C.2). On the other hand, representation learning through TD losses, as in TD-JEPA, is largely understudied. The closest studies (Blier et al., 2021; Lan et al., 2023) show that, under certain parameterizations and assumptions, TD representation learning can recover low-rank decompositions of the successor measure, (i.e. it optimizes the corresponding approximation loss). While these works rely on having a single policy, we provide a first result connecting latent-predictive TD learning with TD learning over the successor measure for multiple policies.

|  | Laplacian | ICVF* | HILP | FB | RLDP | BYOL* | BYOL-$\gamma$* | TD-JEPA |
|---|---|---|---|---|---|---|---|---|
| DMC$_{RGB}$ (avg) | 293.1 ±15.1 | 438.7 ±14.9 | 391.2 ±23.8 | 456.2 ±8.6 | 525.7 ±13.3 | 513.8 ±11.6 | 582.4 ±9.8 | **628.8 ±5.5** |
| walker | 309.4 ±50.0 | 534.9 ±61.3 | 422.8 ±32.5 | 324.4 ±16.6 | 576.1 ±35.3 | 595.2 ±9.0 | 648.3 ±36.5 | **738.9 ±3.5** |
| cheetah | 242.4 ±29.6 | 394.9 ±30.1 | 333.0 ±86.6 | 622.4 ±23.1 | 605.3 ±23.5 | 468.0 ±46.7 | 679.8 ±17.1 | **706.0 ±4.1** |
| quadruped | 430.1 ±32.3 | 583.3 ±17.2 | 513.9 ±10.8 | 475.4 ±16.7 | 551.1 ±23.4 | 581.8 ±16.6 | 570.0 ±6.6 | **626.7 ±13.6** |
| pointmass | 190.4 ±12.4 | 241.6 ±35.6 | 294.9 ±33.4 | 402.8 ±16.8 | 370.3 ±12.0 | 410.3 ±8.5 | **431.6 ±17.4** | **443.7 ±10.9** |
| DMC (avg) | 591.1 ±10.7 | 619.3 ±10.3 | 620.1 ±8.4 | 648.2 ±4.1 | 610.2 ±13.5 | 618.6 ±10.5 | **645.4 ±10.5** | **661.2 ±6.3** |
| walker | 769.7 ±4.7 | 727.0 ±16.2 | 796.4 ±7.7 | **811.5 ±5.9** | 723.9 ±18.3 | 746.8 ±11.0 | 786.1 ±9.6 | 785.2 ±6.7 |
| cheetah | 614.5 ±18.9 | 606.3 ±16.8 | 618.3 ±5.8 | 672.7 ±4.9 | 575.6 ±44.9 | 622.8 ±23.9 | 647.2 ±9.0 | **688.7 ±6.7** |
| quadruped | 635.0 ±38.7 | **708.5 ±14.2** | **694.8 ±11.0** | 595.6 ±9.1 | 665.0 ±13.9 | 611.8 ±28.1 | **683.1 ±26.1** | **691.4 ±5.0** |
| pointmass | 345.1 ±22.4 | 435.5 ±11.1 | 371.0 ±37.1 | **513.0 ±20.0** | 476.3 ±39.4 | 493.0 ±41.3 | 465.1 ±17.6 | 479.3 ±23.6 |
| OGBench$_{RGB}$ (avg) | 30.58 ±0.81 | 25.22 ±0.55 | 32.56 ±0.92 | 39.89 ±0.47 | 39.09 ±0.59 | 40.33 ±0.52 | **41.58 ±0.64** | **41.34 ±0.45** |
| antmaze-mn | 92.20 ±2.91 | 85.80 ±3.02 | 84.60 ±3.59 | **96.80 ±0.74** | **97.60 ±0.50** | 94.40 ±1.48 | **98.00 ±0.73** | 96.67 ±1.11 |
| antmaze-ln | 35.40 ±2.97 | 42.60 ±2.84 | 47.00 ±4.04 | **76.80 ±2.33** | 63.60 ±3.89 | 62.20 ±3.42 | 68.80 ±2.70 | 74.60 ±3.35 |
| antmaze-ms | 60.20 ±3.88 | 46.20 ±2.74 | 71.80 ±2.22 | 86.20 ±2.05 | **90.60 ±1.91** | **90.40 ±1.97** | 86.00 ±3.10 | 84.40 ±3.85 |
| antmaze-ls | 7.20 ±1.98 | 7.20 ±1.20 | 23.60 ±1.83 | 27.40 ±2.78 | 21.80 ±1.01 | 26.60 ±2.23 | 28.60 ±1.71 | **28.80 ±2.50** |
| antmaze-me | 0.00 ±0.00 | 0.00 ±0.00 | 0.20 ±0.20 | **1.80 ±1.09** | 0.80 ±0.44 | **1.20 ±1.00** | **3.20 ±1.98** | 0.20 ±0.20 |
| cube-single | **73.80 ±3.53** | 34.80 ±7.03 | 56.40 ±3.82 | 62.00 ±2.27 | 63.20 ±3.91 | **75.40 ±2.58** | 76.40 ±3.24 | 67.80 ±3.67 |
| cube-double | **1.60 ±0.72** | 0.80 ±0.44 | **1.60 ±0.58** | 1.20 ±0.61 | **2.20 ±1.31** | **2.40 ±0.65** | 1.40 ±0.67 | **3.00 ±0.91** |
| scene | 2.80 ±1.12 | 8.40 ±1.45 | 5.40 ±1.63 | 4.20 ±0.87 | 9.40 ±1.33 | 8.80 ±1.64 | **11.20 ±1.82** | **14.20 ±2.22** |
| puzzle-3x3 | **2.00 ±1.40** | 1.20 ±0.44 | **2.44 ±0.99** | **2.60 ±0.79** | **2.60 ±0.79** | 1.60 ±0.40 | 0.60 ±0.31 | **2.40 ±0.83** |
| OGBench (avg) | 14.81 ±1.32 | 30.87 ±0.58 | **37.98 ±1.11** | **39.04 ±0.66** | 27.07 ±0.83 | 26.42 ±0.83 | 30.42 ±0.94 | **37.98 ±0.77** |
| antmaze-mn | 50.00 ±4.94 | **79.80 ±2.62** | **83.60 ±2.63** | 73.00 ±2.72 | 74.60 ±4.15 | 58.40 ±2.00 | 51.40 ±1.55 | 70.40 ±3.72 |
| antmaze-ln | 21.60 ±3.90 | **58.40 ±1.90** | 52.60 ±3.86 | 36.80 ±4.28 | 36.40 ±4.66 | 26.60 ±3.03 | 21.80 ±3.57 | **57.20 ±4.25** |
| antmaze-ms | 21.40 ±4.32 | 30.90 ±3.30 | 50.60 ±2.46 | **70.40 ±3.95** | 58.40 ±3.29 | 60.60 ±5.07 | 45.60 ±2.84 | 61.56 ±4.53 |
| antmaze-ls | 11.80 ±1.47 | 13.20 ±1.64 | 12.20 ±1.75 | **49.80 ±5.64** | 19.60 ±2.73 | 25.80 ±4.28 | 20.20 ±1.80 | 40.60 ±2.51 |
| antmaze-me | 0.80 ±0.61 | 0.00 ±0.00 | 2.00 ±0.84 | **51.60 ±2.65** | 4.80 ±2.35 | 11.40 ±2.29 | 19.60 ±2.53 | 20.20 ±2.39 |
| cube-single | 15.11 ±1.49 | 20.40 ±1.93 | 74.20 ±3.53 | 49.60 ±3.83 | 19.80 ±2.41 | 22.00 ±3.16 | **79.40 ±2.83** | 34.20 ±2.88 |
| cube-double | 2.00 ±0.42 | 5.00 ±0.80 | **20.00 ±2.72** | 2.60 ±0.43 | 3.80 ±0.76 | 4.40 ±0.72 | 2.60 ±0.67 | 3.60 ±0.78 |
| scene | 7.80 ±1.28 | **45.40 ±2.29** | **43.80 ±1.90** | 12.80 ±1.61 | 11.60 ±1.57 | 15.40 ±1.37 | 14.40 ±2.32 | 38.44 ±1.37 |
| puzzle-3x3 | 2.80 ±0.68 | 16.60 ±0.73 | 2.80 ±0.68 | 4.80 ±0.68 | 14.60 ±0.90 | 13.20 ±1.91 | **18.80 ±0.44** | 15.60 ±1.11 |

Table 1: Performance of zero-shot algorithms for DMC (reward) and OGBench (success rate) with either proprioception or RGB inputs. We report means and standard errors across seeds. Numbers are bold for top algorithms if confidence intervals overlap.

## 6 EXPERIMENTS

We benchmark zero-shot performance across a diverse set of problems, including 4 locomotion/navigation domains from ExoRL/DMC (Tassa et al., 2018; Yarats et al., 2022a), as well as 9 navigation/-manipulation domains from OGBench (Park et al., 2025a). The former suite involves reward-based tasks and high-coverage data, while the latter evaluates goal-reaching and provides low-coverage datasets[4]. We consider both proprioceptive and pixel-based variants of all domains, and report expected returns/success rates across a set of tasks (4-8 depending on the domain) as main evaluation metric. In DMC, we often normalize returns by the maximum achievable (1000).

We structure our evaluation in four parts: **(i)** a comprehensive evaluation of TD-JEPA with respect to existing zero-shot methods; **(ii)** an ablation over the prediction target, measuring the impact of multi-step, policy-aware dynamics modeling; **(iii)** a comparison of TD-JEPA to its symmetric variant that learns a shared state-task encoder $\phi$; and **(iv)** a demonstration of fast adaptation from pre-trained state representations. Further results are presented in App. D, and implementation details in App. E.

**How does TD-JEPA compare to zero-shot RL algorithms?** We first compare TD-JEPA to three groups of successor-feature-based zero-shot RL baselines:[5]

- *Laplacian* (Wu et al., 2019), *HILP* (Park et al., 2024), and *FB* (Touati & Ollivier, 2021) are established zero-shot methods that train a task encoder $\psi$, without specific learning objectives for a state encoder.
- *BYOL*[⋆] (Grill et al., 2020), *BYOL-$\gamma$*[⋆] (Lawson et al., 2025) and *RLDP* (Jajoo et al., 2025) learn a state encoder $\phi$ via latent-predictive learning, which we then use as a task encoder for successor features (learned through a contrastive loss in the case of RLDP).
- *ICVF*[⋆] (Ghosh et al., 2023) learns a multilinear decomposition of the successor measure via expectile regression, yielding both state and task encoders on top of which we train successor features.

For a fair comparison, each method is tuned over comparable hyperparameter grids and adopts the same architecture: in particular, the state input is always passed through an explicit state encoder

---

[4]We additionally apply BC regularization in OGBench based on Park et al. (2025b), as detailed in App. E.6

[5]Notice that only *Laplacian*, *HILP*, *FB* and *RLDP* are standard zero-shot unsupervised RL algorithms, while *BYOL*, *BYOL-$\gamma$*, and *ICVF* (henceforth marked with a ∗) are representation learning methods: their instantiation in a zero-shot framework is novel and designed to investigate the impact of different representations.

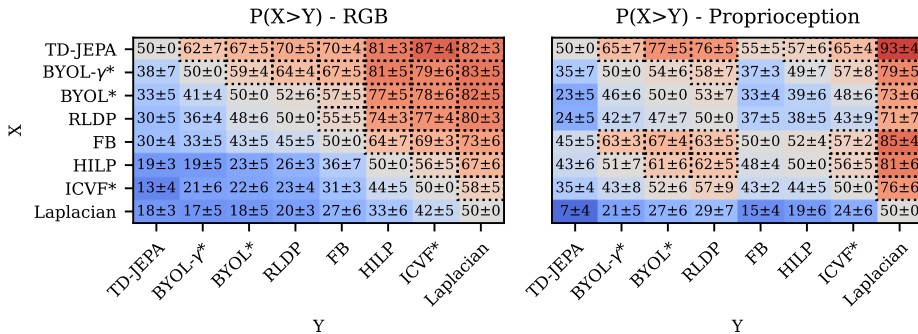

Figure 2: Probabilities of improvement: how likely is method X to outperform method Y on a random domain? We report symmetrized 95% simple bootstrap confidence intervals. Dotted lines surround matches in which the improvement is statistically significant.

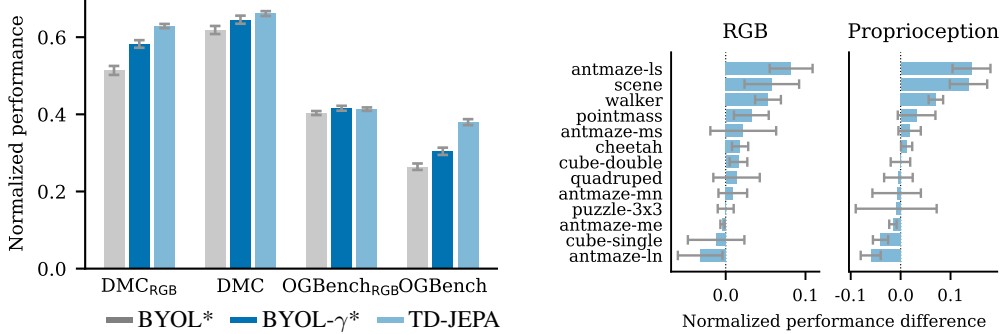

Figure 3: **Left:** normalized zero-shot performance for latent-predictive methods. **Right:** difference in normalized performance between TD-JEPA and its symmetric variant. Error bars represent standard errors on normalized performance or its differences, respectively.

before being fed into, e.g., the successor features estimator $F(s, a; z)$[6]. We find that this protocol results in significant improvements in zero-shot performances, even for existing methods (e.g., $1.3\times$ and $2.4\times$ higher than overlapping pixel-based results for the methods presented in Park et al. (2024) and Jajoo et al. (2025), respectively), as displayed in Tab 1. When considering suite-aggregated performance, we find that TD-JEPA is on par or better than the best performing baseline in each suite. Given the diverse nature of suites (proprioception vs pixels), domains (locomotion, navigation, manipulation) and datasets (high- vs low-coverage), many algorithms unsurprisingly achieve strong performance in some configurations while under-performing in others. We thus additionally measure how consistently well each algorithm performs by computing the probability of improvement (Agarwal et al., 2021) across all domains in Fig. 2. We find that TD-JEPA is consistently among the top performing algorithms, whereas most baselines perform well on a narrow subset of problems. For instance, while TD-JEPA is only slightly preferable to FB and HILP from proprioception, it is significantly better than them in visual domains. Similarly, BYOL-$\gamma$ is slightly better than TD-JEPA in OGBench$_{RGB}$, but it is significantly worse in DMC$_{RGB}$ and OGBench. Finally, we note that latent-predictive methods tend to be generally preferrable in pixel-based domains.

**Which dynamics should latent-predictive zero-shot algorithms model?** The baselines based on BYOL and BYOL-$\gamma$ are algorithmically closest to TD-JEPA, and allow a precise investigation on the dynamics to model. While BYOL$^\star$ and BYOL-$\gamma^\star$ approximate one-step and multi-step transitions of the behavioral policy, respectively, TD-JEPA models multi-step transitions *of the zero-shot policies*. While approximating the behavioral dynamics can be effective for expert-like data (i.e., in OGBench), we observe a general pattern suggesting that directly modeling policy-conditional successor measures is on average beneficial, as reported in Fig. 3 (*left*).

**Should state and task representations differ?** TD-JEPA trains separate state and task encoders: while this may grant a better approximation of successor measures, sharing state and task representations while optimizing a single objective (see Section 3.1) may in practice be more efficient. We measure the difference in per-task normalized performance between TD-JEPA and a symmetric

---

[6]On average, explicit state encoders actually improve the performance for existing methods, see App. D.1.

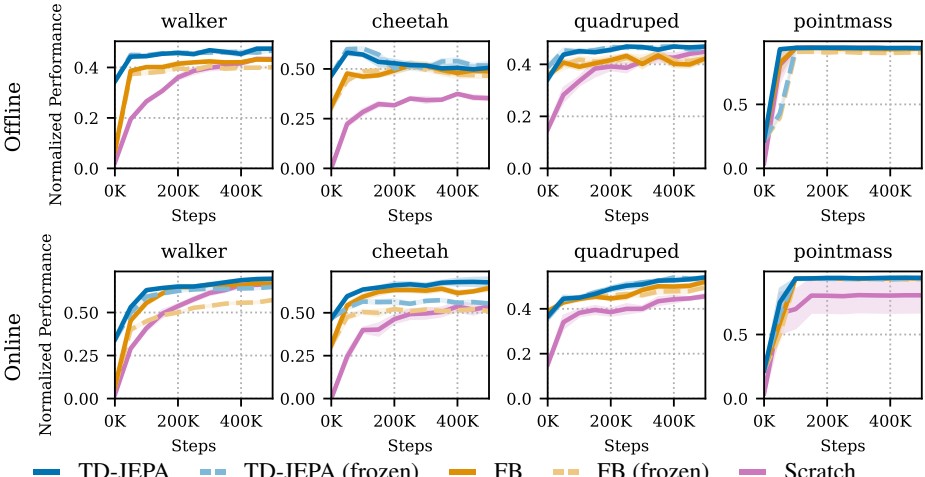

Figure 4: Normalized performance of zero-shot policies when fine-tuned offline (top) or online (bottom). Initializing the agent to zero-shot solutions (blue and yellow lines) results in sample-efficient learning; frozen representations (dashed) are often expressive enough to enable fast adaptation.

variant in Figure 3 (*right*): we observe that this variant performs comparatively rather well, while relying on a single predictor-encoder pair. However, using distinct state and task embeddings tends to improve empirical performance more often than not.

**Are state representations beneficial for fast adaptation?** While the previous evaluations have focused on aggregated zero-shot performance, we now investigate an additional benefit of explicit state representations: fast adaptation at test-time. Given a pixel-based task, we initialize the agent with the zero-shot policy $\pi_z$ and critic learned at pre-training, and we either *fine-tune* the whole model via TD3 (Fujimoto et al., 2018) or keep the pre-trained state encoder *frozen*. We consider two RL adaptation protocols (i) **Offline**: a transition-reward dataset is provided $\mathcal{D}_{\text{rew}} = \{(s, a, s', r)\}$ and TD3 updates are applied offline; (ii) **Online**: an online buffer is additionally collected over time and batches are sampled by mixing it with the offline buffer mentioned above (following the unsupervised-to-online protocol of Kim et al. (2024)). Figure 4 reports results for each DMC domain for the task in which the gap between online and zero-shot algorithms is largest; we consider TD-JEPA and FB as strong, representative algorithms among self-predictive and contrastive methods. We first observe that fine-tuning pre-trained agents leads to large gains in sample efficiency w.r.t. training from scratch, and reaches the asymptotic performance of TD3. More interestingly, frozen representations are often sufficient for downstream learning, and do not need further fine-tuning. We refer to App. D.3 and App. E.7 for further results and details, respectively.

## 7 CONCLUSION

Through the introduction of a novel temporal-difference latent-predictive loss, we presented a zero-shot unsupervised RL method that operates entirely in latent space and can be shown to recover a factorization of the successor measures of multiple policies. Our method tackles a fundamental representation learning problem for control, and highlights a connection between downstream performance and accurate modeling of the successor measure. We thus suggest that flexible representations for RL, particularly for value estimation and optimization on downstream tasks, should be predictive of future behaviors, and precisely capture their *diverse* and *long-term* nature. Empirically, we found that TD-JEPA matches the best zero-shot methods when learning from proprioception, and exceeds them when learning from pixels, while also retrieving state representations that allow fast downstream adaptation. As formal guarantees rely on an assumption of symmetry, one exciting direction for future work may study learning objectives that are compatible with asymmetric successor measures, yet remain amenable to practical optimization. On a practical note, we believe that benchmarking latent-predictive zero-shot objectives on large-scale, real robotic dataset can shed further light on opportunities and limitations of this promising framework.

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

## A  EXTENDED RELATED WORK

As TD-JEPA bridges zero-shot reinforcement learning and latent-predictive representation learning, we reserve this section to building connections to several related works in both areas beyond what was discussed in Sec. 1 and 5.

**Other representation learning methods for RL**  Beyond latent-predictive methods, ICVF (Ghosh et al., 2023; Bhateja et al., 2023) has also been proposed as a multi-policy, multi-step representation learning objective. By relying on an implicit, value-based loss, this method may be applied to off-policy, action-free, transition-based data to recover a decomposition of policy-dependent successor measures. This decomposition is multi-linear, but, unlike TD-JEPA, it is restricted to linear predictors; its practical implementation moreover forces the same dimensionality across state and task embeddings. As BYOL-$\gamma$, ICVF does not natively support zero-shot RL: we thus integrate it into a successor-features-based zero-shot policy optimization scheme.

VIP (Ma et al., 2023) also works in a similar setting as ICVF. It casts representation learning as an offline goal-conditioned reinforcement learning problem. This can be seen related to approximating the successor measure of several goal-reaching policies. On the other hand, TD-JEPA does so for all reward-maximing policies, hence encompassing the goal-reaching ones.

MR.Q (Fujimoto et al., 2025) learns model-based representations that approximately linearize value functions for reward-based RL. This is achieved by combining reward prediction with a single-step latent dynamics loss similar to the one of BYOL, yielding state encoders on top of which value functions and policies are trained. TD-JEPA also aims at linearizing value functions, but does so with a multi-step policy-dependent loss and, most importantly, in a reward-free manner.

More broadly, several recent works evaluate visual representations pre-trained from large-scale data (e.g., internet videos) in control problems (Parisi et al., 2022; Majumdar et al., 2023; Silwal et al., 2024; Schneider et al., 2024; Zhou et al., 2024; McCarthy et al., 2025; Tsagkas et al., 2025). While there is no "best" method overall, as performance are problem and data dependent (Majumdar et al., 2023), representations pre-trained or fine-tuned with RL-related objectives, like time and task awareness (Ma et al., 2023; Bhateja et al., 2023; Tsagkas et al., 2025), perform well in general. TD-JEPA aligns with this view, as it shows that visual representations pre-trained with multi-step and policy-dependent objectives are suitable for value estimation and optimization across multiple tasks.

**Applications of Unsupervised RL**  Training on large, reward-free, multi-task offline datasets, which is an important motivation for this work, is progressively demonstrating its relevance for real-world applications. Current efforts in robotics control, and in particular manipulation, have identified in-the-wild videos as a promising source of information to tap into. These data sources typically include demonstrations of several tasks, and are largely reward-free (Khazatsky et al., 2025; Open X-embodiment collaboration et al., 2024). Recent works have pushed this direction significantly, either by leveraging action annotation (Sobal et al., 2025; Assran et al., 2025b) or latent actions (Ye et al., 2025). Other works in this space focus on large teleoperation datasets, which are again multi-task and reward-free. This is particularly the case of modern VLAs (Team et al., 2025), which have achieved impressive applications in the real world.

Important developments have also taken place in whole-body humanoid control. In this area, agents are normally trained on motion-capture datasets, which can be regarded as multi-task unlabeled data, with recent works achieving impressive results on real hardware (Zeng et al., 2025; He et al., 2025; Weng et al., 2025). The main limitation of these methods is that they focus exclusively on imitating the behaviors in the data (e.g., via motion tracking). Unsupervised zero-shot RL methods have recently stepped in as a promising solution, enabling agents that can quickly solve tasks beyond imitation, like reward optimization and goal reaching. They have been successfully applied in both simulation (Tirinzoni et al., 2025) and on real hardware (Li et al., 2025).

# B PROOFS

## B.1 PROOF OF PROPOSITION 1

For any $z, s, a, s^+$, the term inside the expectation in Eq. 5 can be rewritten as

$$\|T_\phi(\phi(s), a, z) - \overline{\phi(s^+)}\|^2 = \|T_\phi(\phi(s), a, z) \pm \overline{F_\phi^{\pi_z}(s,a)} - \overline{\phi(s^+)}\|^2$$
$$= \|T_\phi(\phi(s), a, z) - \overline{F_\phi^{\pi_z}(s,a)}\|^2 + \|\overline{F_\phi^{\pi_z}(s,a)} - \overline{\phi(s^+)}\|^2$$
$$- (T_\phi(\phi(s), a, z) - \overline{F_\phi^{\pi_z}(s,a)})^\mathsf{T}(\overline{F_\phi^{\pi_z}(s,a)} - \overline{\phi(s^+)}).$$

Taking the expectation w.r.t. $s^+ \sim M^{\pi_z}(\cdot|s,a)$, it is easy to see that the last term is zero since $F_\phi^{\pi_z}(s,a) := \mathbb{E}_{s^+ \sim M^{\pi_z}(\cdot|s,a)}[\phi(s^+)]$, while the second term is a constant. This concludes the proof.

$\square$

**Remark 1.** *By replacing the target $\overline{\phi(s^+)}$ above with $\overline{\psi(s^+)}$ and $F_\phi^{\pi_z}(s,a)$ with $F_\psi^{\pi_z}(s,a)$, this proof generalizes to the Monte Carlo loss with asymmetric representations (Eq. 8).*

## B.2 PROOF OF THEOREM 1

We begin by rewriting the MC-JEPA loss of Eq. 8 with the notation of Sec. 4 as

$$\mathcal{L}_{\text{MC-JEPA}}(\phi, T_{\phi,z}, \psi) := \frac{1}{2}\mathbb{E}_{z \sim \mathcal{Z}, s \sim \rho, s^+ \sim M^{\pi_z}(\cdot|s)}\left[\left\|T_{\phi,z}^\mathsf{T}\phi(s) - \overline{\psi(s^+)}\right\|_2^2\right], \quad (13)$$

where $\rho$ denotes the state distribution. Proposition 1 implies that $\mathcal{L}_{\text{MC-JEPA}}$ has the same gradients as

$$\mathcal{L}_{\text{mc}}(\phi, T_{\phi,z}, \psi) := \frac{1}{2}\mathbb{E}_{z \sim \mathcal{Z}, s \sim \rho}\left[\left\|T_{\phi,z}^\mathsf{T}\phi(s) - \mathbb{E}_{s^+ \sim M^{\pi_z}(\cdot|s)}\left[\overline{\psi(s^+)}\right]\right\|_2^2\right].$$

Let $D_\rho \in \mathbb{R}^{S \times S}$ be a diagonal matrix containing $\rho(s)$ for all states $s \in \mathcal{S}$ on its diagonal. Using that $D_\rho = I$ by Assumption A2,[7]

$$\mathcal{L}_{\text{mc}}(\phi, T_z, \psi) := \frac{1}{2}\mathbb{E}_{z \sim \mathcal{Z}}\left[\|D_\rho^{1/2}(\phi T_z - \overline{M^{\pi_z}\psi})\|_F^2\right] = \frac{1}{2}\mathbb{E}_{z \sim \mathcal{Z}}\left[\|\phi T_z - \overline{M^{\pi_z}\psi}\|_F^2\right]. \quad (14)$$

We now prove all statements for $\mathcal{L}_{\text{mc}}$, as gradient equivalence with Eq. 13 implies they also hold for $\mathcal{L}_{\text{MC-JEPA}}$.

**Statement 1** Let us first compute the optimal predictors. For any $z \in \mathcal{Z}$, the gradient of $\mathcal{L}_{\text{mc}}(\phi, T_z, \psi)$ w.r.t. $T_z$ is

$$\nabla_{T_z}\mathcal{L}_{\text{mc}}(\phi, T_z, \psi) = p(z)\phi^\mathsf{T}(\phi T_z - M^{\pi_z}\psi) = p(z)(T_z - \phi^\mathsf{T}M^{\pi_z}\psi),$$

where $p(z)$ is the probability to sample $z$[8], while the second equality uses that $\phi^\mathsf{T}\phi = I$ by Assumption A1. This yields $T_{\phi,z}^\star = \phi^\mathsf{T}M^{\pi_z}\psi$. Moreover, by simply inverting the roles of $\phi$ and $\psi$, we find that $\nabla_{T_z}\mathcal{L}_{\text{mc}}(\psi, T_z, \phi) = p(z)(T_z - \psi^\mathsf{T}M^{\pi_z}\phi)$ and, thus, $T_{\psi,z}^\star = \psi^\mathsf{T}M^{\pi_z}\phi$.

The gradient of $\mathcal{L}_{\text{SM}}(\phi, T_z, \psi)$ w.r.t. $T_z$ is

$$\nabla_{T_z}\mathcal{L}_{\text{SM}}(\phi, T_z, \psi) = p(z)\phi^\mathsf{T}(\phi T_z \psi^\mathsf{T} - M^{\pi_z})\psi = p(z)(T_z - \phi^\mathsf{T}M^{\pi_z}\psi),$$

where we used again Assumption A1. Hence, $T_z^\star = \phi^\mathsf{T}M^{\pi_z}\psi$. Therefore, we clearly have that $T_z^\star = T_{\phi,z}^\star$. Moreover, since $P^{\pi_z}$ is symmetric by Assumption A3, $M^{\pi_z}$ is symmetric too, and

$$T_{\psi,z}^\star = (\phi^\mathsf{T}(M^{\pi_z})^\mathsf{T}\psi)^\mathsf{T} = (\phi^\mathsf{T}M^{\pi_z}\psi)^\mathsf{T} = (T_z^\star)^\mathsf{T}.$$

Finally, it is easy to see that $\phi T_{\phi,z}^\star$ and $\psi T_{\psi,z}^\star$ satisfy the stated expressions for $\Pi_\phi := \phi\phi^\mathsf{T}$ and $\Pi_\psi := \psi\psi^\mathsf{T}$, respectively. Moreover, $\Pi_\phi$ and $\Pi_\psi$ are symmetric and idempotent ($\Pi_\phi\Pi_\phi = \Pi_\phi$), hence they are orthogonal projection matrices. This proves the first part of the statement.

---

[7]We ignore the $1/S$ scaling that only multiplies the loss by a constant.

[8]Without loss of generality, we also assume that $p(z) > 0$ for all $z$. If this is not the case, any $z$ with $p(z) = 0$ can be removed from the loss.

**Statement 2** Let us now fix any $T_z$ for all $z \in \mathcal{Z}$ and compute the gradients w.r.t. $\phi$ and $\psi$.

$$\nabla_\phi \mathcal{L}_{\mathrm{mc}}(\phi, T_z, \psi) = \mathbb{E}_{z \sim \mathcal{Z}} \big[ (\phi T_z - M^{\pi_z} \psi) T_z^\mathsf{T} \big],$$

$$\nabla_\psi \mathcal{L}_{\mathrm{mc}}(\psi, T_z, \phi) = \mathbb{E}_{z \sim \mathcal{Z}} \big[ (\psi T_z - M^{\pi_z} \phi) T_z^\mathsf{T} \big],$$

$$\nabla_\phi \mathcal{L}_{\mathrm{SM}}(\phi, T_z, \psi) = \mathbb{E}_{z \sim \mathcal{Z}} \big[ (\phi T_z \psi^\mathsf{T} - M^{\pi_z}) \psi T_z^\mathsf{T} \big] = \mathbb{E}_{z \sim \mathcal{Z}} \big[ (\phi T_z - M^{\pi_z} \psi) T_z^\mathsf{T} \big],$$

$$\nabla_\psi \mathcal{L}_{\mathrm{SM}}(\phi, T_z^\mathsf{T}, \psi) = \mathbb{E}_{z \sim \mathcal{Z}} \big[ (\psi T_z \phi^\mathsf{T} - (M^{\pi_z})^\mathsf{T}) \phi T_z^\mathsf{T} \big] = \mathbb{E}_{z \sim \mathcal{Z}} \big[ (\psi T_z - (M^{\pi_z})^\mathsf{T} \phi) T_z^\mathsf{T} \big],$$

where we used Assumption A1 to simplify the last two expressions. Given that $P^{\pi_z}$ and, thus, $M^{\pi_z}$ are symmetric by Assumption A3, these gradients match as stated.

$\square$

### B.3 Proof of Theorem 2

We begin by rewriting the TD-JEPA loss of Eq. 9 with the notation of Sec. 4 as

$$\mathcal{L}_{\text{TD-JEPA}}(\phi, T_z, \psi) := \frac{1}{2} \mathbb{E}_{z \sim \mathcal{Z}, s \sim \rho, s^+ \sim P^{\pi_z}(\cdot|s)} \left[ \left\| T_{\phi,z}^\mathsf{T} \phi(s) - \overline{\psi(s^+)} - \gamma \overline{T_{\phi,z}^\mathsf{T} \phi(s^+)} \right\|_2^2 \right]. \tag{15}$$

Following the proof of Theorem 1, we can put this in matrix form as

$$\mathcal{L}_{\mathrm{td}}(\phi, T_z, \psi) := \frac{1}{2} \mathbb{E}_{z \sim \mathcal{Z}} \left[ \left\| D_\rho^{1/2} (\phi T_z - U_z) \right\|_F^2 \right], \tag{16}$$

where $U_z := \overline{P^{\pi_z} \psi} - \gamma \overline{P^{\pi_z} \phi T_z}$. As we only brought expectations inside the norm, Eq. 16 has the same gradients as Eq. 15. Hence, we define a continuous-time relaxation of gradient descend dynamics for Eq. 16 (equiv. Eq. 15) by the following ordinary differential equation (ODE) system:

$$\begin{cases} T_{\phi,z,t} \in \arg\min_{T_z} \mathcal{L}_{\mathrm{td}}(\phi_t, T_z, \psi_t) \\ T_{\psi,z,t} \in \arg\min_{T_z} \mathcal{L}_{\mathrm{td}}(\psi_t, T_z, \phi_t) \\ \dot{\phi}_t = -\nabla_{\phi_t} \mathcal{L}_{\mathrm{td}}(\phi_t, T_{\phi,z,t}, \psi_t) \\ \dot{\psi}_t = -\nabla_{\psi_t} \mathcal{L}_{\mathrm{td}}(\psi_t, T_{\psi,z,t}, \phi_t) \end{cases} \tag{17}$$

This implicitly assumes that predictors are optimized at a much faster rate than representations – an important property used in Theorem 1 of Tang et al. (2023) to show constant covariance and, thus, no collapse. We now prove this by following similar steps as in the proof of Tang et al. (2023), adapted to our setting. In particular, we prove it for the representations $\phi$ only. Given the symmetry of the losses, the same result can trivially be proven for $\psi$ as well.

We need to prove that $\frac{\mathrm{d}}{\mathrm{d}t}(\phi_t^\mathsf{T} \phi_t) = 0$. Since $\frac{\mathrm{d}}{\mathrm{d}t}(\phi_t^\mathsf{T} \phi_t) = (\phi_t^\mathsf{T} \dot{\phi}_t)^\mathsf{T} + \phi_t^\mathsf{T} \dot{\phi}_t$, it is enough to show that $\phi_t^\mathsf{T} \dot{\phi}_t = 0$. Simple algebra yields

$$\nabla_\phi \mathcal{L}_{\mathrm{td}}(\phi, T_z, \psi) = \mathbb{E}_z \left[ D_\rho (\phi T_z - U_z) T_z^\mathsf{T} \right],$$

$$\nabla_{T_z} \mathcal{L}_{\mathrm{td}}(\phi, T_z, \psi) = p(z) \phi^\mathsf{T} D_\rho (\phi T_z - U_z).$$

Therefore,

$$\begin{aligned} \phi_t^\mathsf{T} \dot{\phi}_t &= -\phi_t^\mathsf{T} \nabla_{\phi_t} \mathcal{L}_{\mathrm{td}}(\phi_t, T_{\phi,z,t}, \psi_t) \\ &= -\phi_t^\mathsf{T} \mathbb{E}_z \left[ D_\rho (\phi_t T_{\phi,z,t} - U_z) T_{\phi,z,t}^\mathsf{T} \right] \\ &= -\sum_{z \in \mathcal{Z}} p(z) \phi_t^\mathsf{T} D_\rho (\phi_t T_{\phi,z,t} - U_z) T_{\phi,z,t}^\mathsf{T} \\ &= -\sum_{z \in \mathcal{Z}} \nabla_{T_{\phi,z,t}} \mathcal{L}_{\mathrm{td}}(\phi_t, T_{\phi,z,t}, \psi_t) T_{\phi,z,t}^\mathsf{T} \\ &= 0, \end{aligned}$$

where the last equation holds since the gradient w.r.t. the predictor is zero at every step (first order optimality conditions from Eq. 17).

### B.4 PROOF OF THEOREM 3

We begin by rewriting the TD-JEPA loss of Eq. 9 with the notation of Sec. 4 as

$$\mathcal{L}_{\text{TD-JEPA}}(\phi, T_z, \psi) := \frac{1}{2}\mathbb{E}_{z\sim\mathcal{Z}, s\sim\rho, s^+\sim P^{\pi_z}(\cdot|s)}\left[\left\|\left\|T_{\phi,z}^\mathsf{T}\phi(s) - \overline{\psi(s^+)} - \gamma\overline{T_{\phi,z}^\mathsf{T}\phi(s^+)}\right\|\right\|_2^2\right]. \quad (18)$$

Following the proof of Theorem 1, we can put this in matrix form as

$$\mathcal{L}_{\text{td}}(\phi, T_z, \psi) := \frac{1}{2}\mathbb{E}_{z\sim\mathcal{Z}}\left[\left\|\left\|D_\rho^{1/2}\left(\phi T_z - \overline{P^{\pi_z}\psi} - \gamma\overline{P^{\pi_z}\phi T_z}\right)\right\|\right\|_F^2\right]. \quad (19)$$

As we only brought expectations inside the norm, Eq. 19 has the same gradients as Eq. 18, so we can focus on it to prove the results. Moreover, we can set $D_\rho = I$ by Assumption A2.

**Statement 1** We start by computing the gradients of $\mathcal{L}_{\text{td}}$ and $\mathcal{L}_{\text{fw}}$ w.r.t. $T_z$. Up to a multiplicative constant $p(z)$ (which doesn't change the results), we have

$$\nabla_{T_z}\mathcal{L}_{\text{td}}(\phi, T_z, \psi) = \phi^\mathsf{T}(\phi T_z - P^{\pi_z}\psi - \gamma P^{\pi_z}\phi T_z) = T_z - \phi^\mathsf{T} P^{\pi_z}\psi - \gamma\phi^\mathsf{T} P^{\pi_z}\phi T_z,$$

where we used Assumption A1 to set $\phi^\mathsf{T}\phi = I$. Further using that $\psi^\mathsf{T}\psi = I$,

$$\begin{aligned}\nabla_{T_z}\mathcal{L}_{\text{fw}}(\phi, T_z, \psi) &= \phi^\mathsf{T}(\phi T_z\psi^\mathsf{T} - P^{\pi_z} - \gamma P^{\pi_z}\phi T_z\psi^\mathsf{T})\psi \\ &= T_z - \phi^\mathsf{T} P^{\pi_z}\psi - \gamma\phi^\mathsf{T} P^{\pi_z}\phi T_z \\ &= \nabla_{T_z}\mathcal{L}_{\text{td}}(\phi, T_z, \psi).\end{aligned}$$

Therefore, the gradients w.r.t. $T_z$ of the two losses match, which means that the stationary points (i.e., optimal predictors) are also the same. Setting these gradients to zero we thus find that

$$T_{\phi,z}^\star = T_{z,\text{fw}}^\star = (\phi^\mathsf{T}(I - \gamma P^{\pi_z})\phi)^{-1}\phi^\mathsf{T} P^{\pi_z}\psi.$$

Note that matrix $\phi^\mathsf{T}(I - \gamma P^{\pi_z})\phi$ is positive definite and, thus, invertible. This is because $I - \gamma P^{\pi_z}$ is positive definite and $\phi^\mathsf{T}\phi = I$. Using that $M^{\pi_z} = (I - \gamma P^{\pi_z})^{-1}P^{\pi_z}$,

$$\phi T_{\phi,z}^\star = \phi T_{z,\text{fw}}^\star = \underbrace{\phi(\phi^\mathsf{T}(I - \gamma P^{\pi_z})\phi)^{-1}\phi^\mathsf{T}(I - \gamma P^{\pi_z})}_{\tilde{\Pi}_{\phi,z}}M^{\pi_z}\psi,$$

where it is easy to verify that $\tilde{\Pi}_{\phi,z}$ is idempotent ($\tilde{\Pi}_{\phi,z}\tilde{\Pi}_{\phi,z} = \tilde{\Pi}_{\phi,z}$) but not necessarily symmetric, hence an oblique projection as stated.

For the other result in Statement 1, we proceed analogously by first showing that

$$\nabla_{T_z}\mathcal{L}_{\text{td}}(\psi, T_z, \phi) = T_z - \psi^\mathsf{T} P^{\pi_z}\phi - \gamma\psi^\mathsf{T} P^{\pi_z}\psi T_z,$$

$$\nabla_{T_z}\mathcal{L}_{\text{bw}}(\phi, T_z, \psi) = T_z - \psi^\mathsf{T}(P^{\pi_z})^\mathsf{T}\phi - \gamma\psi^\mathsf{T}(P^{\pi_z})^\mathsf{T}\psi T_z = \nabla_{T_z}\mathcal{L}_{\text{td}}(\psi, T_z, \phi),$$

where the last equality is true since $P^{\pi_z}$ is symmetric. Then the result follows as before after equating the gradients to zero, solving for $T_{\psi,z}^\star$, and expressing $\psi T_{\psi,z}^\star$ as a function of $\tilde{\Pi}_{\psi,z}$.

**Statement 2** We show that the gradients w.r.t. $\phi$ and $\psi$ match for any predictor $T_z$

$$\nabla_\phi\mathcal{L}_{\text{td}}(\phi, T_z, \psi) = (\phi T_z - P^{\pi_z}\psi - \gamma P^{\pi_z}\phi T_z)T_z^\mathsf{T},$$

$$\nabla_\phi\mathcal{L}_{\text{fw}}(\phi, T_z, \psi) = \left(\phi T_z\psi^\mathsf{T} - P^{\pi_z} - \gamma P^{\pi_z}\phi T_z\psi^\mathsf{T}\right)\psi T_z^\mathsf{T} = \nabla_\phi\mathcal{L}_{\text{td}}(\phi, T_z, \psi),$$

where we used that $\psi^\mathsf{T}\psi = I$. Similarly,

$$\nabla_\psi\mathcal{L}_{\text{td}}(\psi, T_z, \phi) = (\psi T_z - P^{\pi_z}\phi - \gamma P^{\pi_z}\psi T_z)T_z^\mathsf{T},$$

$$\nabla_\psi\mathcal{L}_{\text{bw}}(\phi, T_z, \psi) = \left(\psi T_z\phi^\mathsf{T} - (P^{\pi_z})^\mathsf{T} - \gamma(P^{\pi_z})^\mathsf{T}\psi T_z\phi^\mathsf{T}\right)\phi T_z^\mathsf{T} = \nabla_\psi\mathcal{L}_{\text{td}}(\psi, T_z, \phi),$$

where we used that $\phi^\mathsf{T}\phi = I$ and $(P^{\pi_z})^\mathsf{T} = P^{\pi_z}$. This proves the statement.

$\square$

## B.5 PROOF OF THEOREM 4

Let us start from the first inequality. Defining $V_r^{\pi_z} \in \mathbb{R}^S$ as a vector containing $V_r^{\pi_z}(s)$ for all states $s \in \mathcal{S}$, we can write the left-hand side for any $r \in \mathbb{R}^S$ as

$$\mathbb{E}_{z \in \mathcal{Z}} \left[ \sum_{s \in \mathcal{S}} \left( V_r^{\pi_z}(s) - \phi(s)^\mathsf{T} T_z \omega_r \right)^2 \right] = \mathbb{E}_{z \in \mathcal{Z}} \| V_r^{\pi_z} - \phi T_z \omega_r \|_2^2.$$

Since $V_r^{\pi_z} = M^{\pi_z} r$ and, by Assumption A1, $\omega_r = \psi^\mathsf{T} r$,

$$\begin{aligned} \mathbb{E}_{z \in \mathcal{Z}} \| V_r^{\pi_z} - \phi T_z \omega_r \|_2^2 = \mathbb{E}_{z \in \mathcal{Z}} \| M^{\pi_z} r - \phi T_z \psi^\mathsf{T} r \|_2^2 \\ \leq \mathbb{E}_{z \in \mathcal{Z}} \| M^{\pi_z} - \phi T_z \psi^\mathsf{T} \|_F^2 \| r \|_2^2 \\ = 2\mathcal{L}_{\mathrm{SM}}(\phi, T_z, \psi) \| r \|_2^2. \end{aligned}$$

The inequality is thus obtained by maximizing both sides over rewards with norm bounded by 1.

Let us now prove the bounds of $\mathcal{L}_{\mathrm{SM}}$ in terms of the Bellman errors $\mathcal{L}_{\mathrm{fw}}$ and $\mathcal{L}_{\mathrm{bw}}$. Let us fix $z \in \mathcal{Z}$ and recall that $M^{\pi_z}$ admits both a "forward" Bellman equation, $M^{\pi_z} = P^{\pi_z} + \gamma P^{\pi_z} M^{\pi_z}$, and a "backward" one, $M^{\pi_z} = P^{\pi_z} + \gamma M^{\pi_z} P^{\pi_z}$ (Blier et al., 2021). This implies that $M^{\pi_z} = (I - \gamma P^{\pi_z})^{-1} P^{\pi_z} = P^{\pi_z} (I - \gamma P^{\pi_z})^{-1}$. Then, for any matrix $M \in \mathbb{R}^{S \times S}$,

$$M - M^{\pi_z} = (I - \gamma P^{\pi_z})^{-1} \left( (I - \gamma P^{\pi_z}) M - P^{\pi_z} \right) = (I - \gamma P^{\pi_z})^{-1} \left( M - P^{\pi_z} - \gamma P^{\pi_z} M \right),$$

$$M - M^{\pi_z} = \left( M(I - \gamma P^{\pi_z}) - P^{\pi_z} \right) (I - \gamma P^{\pi_z})^{-1} = \left( M - P^{\pi_z} - \gamma M P^{\pi_z} \right) (I - \gamma P^{\pi_z})^{-1}.$$

Using the first set of equalities with $M = \phi T_z \psi^\mathsf{T}$, we can easily bound

$$\begin{aligned} \mathcal{L}_{\mathrm{SM}}(\phi, T_z, \psi) = \frac{1}{2} \mathbb{E}_{z \sim \mathcal{Z}} \left[ \| \phi T_z \psi^\mathsf{T} - M^{\pi_z} \|_F^2 \right] \\ \leq \frac{1}{2} \mathbb{E}_{z \sim \mathcal{Z}} \left[ \| (I - \gamma P^{\pi_z})^{-1} \|_2^2 \| \phi T_z \psi^\mathsf{T} - P^{\pi_z} - \gamma P^{\pi_z} \phi T_z \psi^\mathsf{T} \|_F^2 \right], \end{aligned}$$

where we used the inequality $\| XY \|_F^2 \leq \| X \|_2^2 \| Y \|_F^2$ with $\| \cdot \|_2^2$ denoting the operator norm and $\| \cdot \|_F^2$ the frobenius norm. Moreover,

$$\| (I - \gamma P^{\pi_z})^{-1} \|_2^2 = \frac{1}{(1 - \gamma)^2} \| (1 - \gamma)(I - \gamma P^{\pi_z})^{-1} \|_2^2 \leq \frac{S}{(1 - \gamma)^2},$$

where the last inequality holds since $(1 - \gamma)(I - \gamma P^{\pi_z})^{-1}$ is a stochastic matrix. Hence,

$$\mathcal{L}_{\mathrm{SM}}(\phi, T_z, \psi) \leq \frac{S}{(1 - \gamma)^2} \mathcal{L}_{\mathrm{fw}}(\phi, T_z, \psi),$$

which proves the first inequality with $c = \frac{S}{(1-\gamma)^2}$. The second one can be proved analogously.

$\square$

## C THEORETICAL ANALYSIS UNDER RELAXED ASSUMPTIONS

This section describes how the main assumptions used in Section 4 can be removed, namely the uniform state distribution (A1), identity covariances (A2), and symmetric kernels $P^{\pi_z}$ or $M^{\pi_z}$ (A3). We shall derive similar results Th. 1 and Th. 3, but at the price of more complex proofs and notation. We do so through a novel "gradient matching" argument that reduces a general latent-predictive loss to density approximation. As we shall see, this encompasses not only MC-JEPA and TD-JEPA, but also existing methods (Tang et al., 2023; Khetarpal et al., 2025; Lawson et al., 2025).

### C.1 REDUCTION FROM LATENT-PREDICTIVE TO DENSITY-BASED LOSSES

Let $\mathcal{Z}$ be a finite set. For $z \in \mathcal{Z}$, let $\Xi_z \in \mathbb{R}^{S \times S}$ be a generic kernel. For instance, $\Xi_z$ may be the successor measure $M^{\pi_z}$ or the one-step kernel $P^{\pi_z}$, but it is not important at this point. Using the

same notation as Section 4 and Appendix B, consider the following density-based loss:

$$\mathcal{L}_{\text{dens}}(\phi, T_z, \psi) := \frac{1}{2} \mathbb{E}_{z \sim \mathcal{Z}, s \sim \rho, s^+ \sim \rho} \left[ \left( \phi(s)^{\mathsf{T}} T_z \psi(s^+) - \frac{\Xi_z(s^+|s)}{\rho(s^+)} \right)^2 \right]$$

$$= \frac{1}{2} \mathbb{E}_{z \sim \mathcal{Z}} \left[ \| D_\rho^{1/2} (\phi T_z \psi^{\mathsf{T}} - \Xi_z D_\rho^{-1}) D_\rho^{1/2} \|_F^2 \right]. \tag{20}$$

Minimizing this loss over representations $\phi, \psi$ and a collection of predictors $\{T_z\}_{z \in \mathcal{Z}}$ is equivalent to finding the best multilinear approximation to the densities of the $\Xi_z$ w.r.t. the state distribution $\rho$. Note that this is a well-defined loss (i.e., it does not involve stop-gradient operations) and the prediction targets $\Xi_z D_\rho^{-1}$ are not a function of the representations being learned. Our goal is to show that certain latent-predictive dynamics optimize this density-based loss.

For didactic purpose, let us consider two abstract latent-predictive losses $\mathcal{L}_\phi(\phi, T_{\phi,z}, \psi)$ and $\mathcal{L}_\psi(\psi, T_{\psi,z}, \phi)$. We shall specify what these are later. $\mathcal{L}_\phi(\phi, T_{\phi,z}, \psi)$ is optimized over $\phi$ and $T_{\phi,z}$ for all $z$, while $\mathcal{L}_\psi(\psi, T_{\psi,z}, \phi)$ is optimized over $\psi$ and $T_{\psi,z}$ for all $z$. As common in the literature, we study a continuous-time relaxation of gradient descend dynamics by assuming that predictors are optimized at a much faster rate than representations – an important property to ensure that the latter ones do not collapse (Tang et al., 2023). This process can be described by the following ordinary differential equation (ODE) system:

$$\begin{cases} T_{\phi,z,t} \in \arg\min_{T_z} \mathcal{L}_\phi(\phi_t, T_z, \psi_t) \\ T_{\psi,z,t} \in \arg\min_{T_z} \mathcal{L}_\psi(\psi_t, T_z, \phi_t) \\ \dot{\phi}_t = -\nabla_{\phi_t} \mathcal{L}_\phi(\phi_t, T_{\phi,z,t}, \psi_t) \\ \dot{\psi}_t = -\nabla_{\psi_t} \mathcal{L}_\psi(\psi_t, T_{\psi,z,t}, \phi_t) \end{cases} \tag{21}$$

We now ask the question: how should $\mathcal{L}_\phi$ and $\mathcal{L}_\psi$ be defined for the dynamics of Eq. 21 to optimize the density-based loss of Eq. 20? The following result provides and answer.

**Theorem 5.** *Suppose that the gradients of $\mathcal{L}_{\text{dens}}$, $\mathcal{L}_\phi$, and $\mathcal{L}_\psi$ match for all $\phi$, $\psi$, $T_z$, i.e.,*

1. $\nabla_{T_z} \mathcal{L}_{\text{dens}}(\phi, T_z, \psi) = \nabla_{T_z} \mathcal{L}_\phi(\phi, T_z, \psi)$

2. $\nabla_{T_z} \mathcal{L}_{\text{dens}}(\phi, T_z^{\mathsf{T}}, \psi) = \nabla_{T_z} \mathcal{L}_\psi(\psi, T_z, \phi)$

3. $\nabla_\phi \mathcal{L}_{\text{dens}}(\phi, T_z, \psi) = \nabla_\phi \mathcal{L}_\phi(\phi, T_z, \psi)$

4. $\nabla_\psi \mathcal{L}_{\text{dens}}(\phi, T_z^{\mathsf{T}}, \psi) = \nabla_\psi \mathcal{L}_\psi(\psi, T_z, \phi)$

*Then, $\mathcal{L}_{\text{dens}}$ is a Lyapunov function for the ODE of Eq. 21.*

*Proof.* Let $T_{z,t} \in \arg\min_{T_z} \mathcal{L}_{\text{dens}}(\phi_t, T_z, \psi_t)$ be the optimal predictor for the density-based loss given $(\phi_t, \psi_t)$ and define $\mathcal{L}(t) := \mathcal{L}_{\text{dens}}(\phi_t, T_{z,t}, \psi_t)$. We verify that a $\mathcal{L}(t)$, and thus $\mathcal{L}_{\text{dens}}$, is a Lyapunov function for the ODE of Eq. 21. First note that $\mathcal{L}_{\text{dens}}$ is continuous and has continuous first derivates. By the chain rule,

$$\frac{\mathrm{d}}{\mathrm{d}t} \mathcal{L}(t) = \nabla_\phi \mathcal{L}_{\text{dens}}(\phi_t, T_{z,t}, \psi_t) \cdot \dot{\phi}_t + \nabla_\psi \mathcal{L}_{\text{dens}}(\phi_t, T_{z,t}, \psi_t) \cdot \dot{\psi}_t,$$

where the $\cdot$ operation is the dot product of the vectorized matrices. By Eq. 21, $\dot{\phi}_t = -\nabla_\phi \mathcal{L}_\phi(\phi_t, T_{\phi,z,t}, \psi_t)$ and $\dot{\psi}_t = -\nabla_\psi \mathcal{L}_\psi(\psi_t, T_{\psi,z,t}, \phi_t)$. Moreover, assumptions 1 and 2 directly yield $T_{\phi,z,t} = T_{z,t}$ and $T_{\psi,z,t} = T_{z,t}^{\mathsf{T}}$. Plugging these into $\dot{\phi}_t$ and $\dot{\psi}_t$ and using assumptions 3 and 4,

$$\dot{\phi}_t = -\nabla_\phi \mathcal{L}_\phi(\phi_t, T_{z,t}, \psi_t) = -\nabla_\phi \mathcal{L}_{\text{dens}}(\phi_t, T_{z,t}, \psi_t),$$

$$\dot{\psi}_t = -\nabla_\psi \mathcal{L}_\psi(\psi_t, T_{z,t}^{\mathsf{T}}, \phi_t) = -\nabla_\psi \mathcal{L}_{\text{dens}}(\phi_t, T_{z,t}, \psi_t).$$

Thus,

$$\frac{\mathrm{d}}{\mathrm{d}t} \mathcal{L}(t) = -\| \nabla_\phi \mathcal{L}_{\text{dens}}(\phi_t, T_{z,t}, \psi_t) \|_F^2 - \| \nabla_\psi \mathcal{L}_{\text{dens}}(\phi_t, T_{z,t}, \psi_t) \|_F^2 \le 0,$$

Moreover, we clearly have $\frac{\mathrm{d}}{\mathrm{d}t} \mathcal{L}(t) < 0$ if $(\phi_t, \psi_t)$ is not a stationary point of the ODE, which proves the statement. $\qquad \square$

Therefore, if the gradients of $\mathcal{L}_{\text{dens}}$, $\mathcal{L}_\phi$, and $\mathcal{L}_\psi$ match, the gradient dynamics of Eq. 21 monotonically improve the density-based loss over time. We remark that this does not imply convergence to the global minimum as $\mathcal{L}_{\text{dens}}$ is not convex. Th. 5 thus suggests a simple trick for designing the right latent-predictive dynamics to optimize a given density-based loss: just find $\mathcal{L}_\phi$ and $\mathcal{L}_\psi$ whose gradients match those of $\mathcal{L}_{\text{dens}}$. Latent predictive losses that satisfy this property are derived in the next result.

**Proposition 2.** *Consider the following latent-predictive losses*

$$\mathcal{L}_\phi(\phi, T_z, \psi) := \frac{1}{2}\mathbb{E}_{z \sim \mathcal{Z}} \|D_\rho^{1/2}(\phi T_z - \overline{\Xi_z \psi \Sigma_\psi^{-1}})\|_{\Sigma_\psi}^2, \tag{22}$$

$$\mathcal{L}_\psi(\psi, T_z, \phi) := \frac{1}{2}\mathbb{E}_{z \sim \mathcal{Z}} \|D_\rho^{1/2}(\psi T_z - \overline{\Xi_z^* \phi \Sigma_\phi^{-1}})\|_{\Sigma_\phi}^2, \tag{23}$$

*where* $\|X\|_W = \|XW^{1/2}\|_F$, $\Sigma_\phi := \phi^\mathsf{T} D_\rho \phi$, $\Sigma_\psi := \psi^\mathsf{T} D_\rho \psi$, *and* $\Xi_z^* := D_\rho^{-1}\Xi_z^\mathsf{T} D_\rho$ *is the* $\rho$-*adjoint of* $\Xi_z$. *Then,* $\mathcal{L}_\phi$ *and* $\mathcal{L}_\psi$ *satisfy the gradient matching conditions for* $\mathcal{L}_{\text{dens}}$ *stated in Th. 5.*

*Proof.* This can be directly obtained through simple linear algebra. $\qquad\square$

Note that, while the latent-predictive losses of Eq. 22 and 23 are expressed in expectation w.r.t. the kernels $\Xi_z$ and $\Xi_z^*$, sample-based estimators for both are possible. In particular, while Eq. 22 involves sampling from the kernel $\Xi_z$, Eq. 23 involves a "backward" sampling operation given by the adjoint of $\Xi_z$.

### C.2 GENERALIZING EXISTING RESULTS

Through the right assumptions and choice of $\Xi_z$, Proposition 2 and Theorem 5 yield several existing results. First, with $\Xi_z = M^{\pi_z}$, we note that Eq. 22 and 23 are equivalent to MC-JEPA (cf. Eq. 8) with additional covariance weighting and a backward-in-time sampling in the loss for $\psi$. Assuming that $D_\rho = I$ (A1), $\Sigma_\phi = \Sigma_\psi = I$ (A2), and $P^{\pi_z}$ is symmetric for all $z$ (A3), this mismatch is resolved and we recover Th. 1 under its same assumptions. On the other hand, Proposition 2 and Theorem 5 show what happens when such assumptions are relaxed: if only A3 holds[9], then Th. 1 still holds up to covariance transformations, while if A3 is further removed then MC-JEPA needs to be modified with backward sampling in the loss of $\psi$ to guarantee optimization of the successor measure approximation loss.

Several analyses in related works, which all assume A1-A3, are also recovered by proper choice of $\Xi_z$. For instance, for a single policy $\pi$ (i.e., $|\mathcal{Z}| = 1$), we obtain Theorem 6 of Tang et al. (2023) by setting $\Xi_z = P^\pi$ and Theorem 4.1 of Lawson et al. (2025) by setting $\Xi_z = M^\pi$. By letting $\mathcal{Z}$ correspond to the action space $\mathcal{A}$, we recover Theorem 2 of Khetarpal et al. (2025) by setting $\Xi_a = P_a^\pi$ for all actions $a$, with $P_a$ the matrix containing $P(s'|s,a)$ for all $(s, s')$.

### C.3 TD-JEPA WITH FORWARD-BACKWARD-IN-TIME SAMPLING

In the previous section, we have seen that MC-JEPA requires a backward-in-time sampling operation in the loss for $\psi$ to provably optimize the successor measure approximation loss. In this section, we derive the TD dynamics corresponding to this process.

We thus start from $\mathcal{L}_{\text{dens}}$, $\mathcal{L}_\phi$, and $\mathcal{L}_\psi$ with $\Xi_z = M^{\pi_z}$ and replace the latter with the one-step kernel $P^{\pi_z}$ plus the bootstrapped parameterization $M^{\pi_z} \simeq \phi T_z \psi^\mathsf{T} D_\rho$. For $\mathcal{L}_{\text{dens}}$, similarly to Th. 3, we derive density-based forward and backward TD losses, respectively based on the application of the forward and backward Bellman operator. For the first one, we use that $M^{\pi_z} = P^{\pi_z} + \gamma P^{\pi_z} M^{\pi_z} \simeq P^{\pi_z} + \gamma P^{\pi_z} \phi T_z \psi^\mathsf{T} D_\rho$ and write

$$\mathcal{L}_{\text{fw}}(\phi, T_z, \psi) := \frac{1}{2}\mathbb{E}_{z \sim \mathcal{Z}}\left[\|D_\rho^{1/2}(\phi T_z \psi^\mathsf{T} - P^{\pi_z}D_\rho^{-1} - \gamma\overline{P^{\pi_z}\phi T_z \psi^\mathsf{T}})D_\rho^{1/2}\|_F^2\right]. \tag{24}$$

---

[9]To be precise, here we need the density $M^{\pi_z}D_\rho^{-1}$ to be symmetric, which implies that $(M^{\pi_z})^* = M^{\pi_z}$.

For the second one, we rewrite the backward Bellman equation in terms of the adjoint of $M^{\pi_z}$ as

$$
\begin{aligned}
(M^{\pi_z})^* &= (P^{\pi_z})^* + \gamma(M^{\pi_z}P^{\pi_z})^* \\
&= (P^{\pi_z})^* + \gamma D_\rho^{-1}(P^{\pi_z})^{\mathsf{T}}(M^{\pi_z})^{\mathsf{T}}D_\rho \\
&\simeq (P^{\pi_z})^* + \gamma D_\rho^{-1}(P^{\pi_z})^{\mathsf{T}}D_\rho\psi T_z^{\mathsf{T}}\phi^{\mathsf{T}}D_\rho \\
&= (P^{\pi_z})^* + \gamma(P^{\pi_z})^*\psi T_z^{\mathsf{T}}\phi^{\mathsf{T}}D_\rho.
\end{aligned}
$$

This yields

$$
\mathcal{L}_{\mathrm{bw}}(\phi, T_z, \psi) := \frac{1}{2}\mathbb{E}_{z\sim\mathcal{Z}}\left[\|D_\rho^{1/2}(\psi T_z\phi^{\mathsf{T}} - (P^{\pi_z})^*D_\rho^{-1} - \gamma(P^{\pi_z})^*\overline{\psi T_z\phi^{\mathsf{T}}})D_\rho^{1/2}\|_F^2\right]. \quad (25)
$$

We then use the same bootstrapping for the latent-predictive losses of Eq. 22 and 23, thus obtaining

$$
\mathcal{L}_\phi(\phi, T_z, \psi) := \frac{1}{2}\mathbb{E}_{z\sim\mathcal{Z}}\|D_\rho^{1/2}(\phi T_z - \overline{P^{\pi_z}\psi\Sigma_\psi^{-1}} - \gamma\overline{P^{\pi_z}\phi T_z})\|_{\overline{\Sigma}_\psi}^2, \quad (26)
$$

$$
\mathcal{L}_\psi(\psi, T_z, \phi) := \frac{1}{2}\mathbb{E}_{z\sim\mathcal{Z}}\|D_\rho^{1/2}(\psi T_z - \overline{(P^{\pi_z})^*\phi\Sigma_\phi^{-1}} - \gamma\overline{(P^{\pi_z})^*\psi T_z})\|_{\overline{\Sigma}_\phi}^2. \quad (27)
$$

As for MC-JEPA, this is the counterpart of TD-JEPA with additional covariance weighting and backward-in-time sampling in the loss of $\psi$. Through simple algebra, it is not difficult to check that $\mathcal{L}_\phi$ and $\mathcal{L}_\psi$ have matching gradients (as in Th. 5) w.r.t. $\mathcal{L}_{\mathrm{fw}}$ and $\mathcal{L}_{\mathrm{bw}}$, respectively. This implies that Th. 3 holds for this modified algorithm without assumptions A1-A3.

Unfortunately, differently from the original TD-JEPA, this variant is not easy to be optimized off-policy. In fact, while for Eq. 26 we can simply replace the on-policy kernel $P^{\pi_z}$ with $P(s'|s, a)$ and condition the predictor on actions, the same trick cannot be used for the adjoint $(P^{\pi_z})^*$ in Eq. 27. This is because there is no action-conditioned backward Bellman equation. We leave the study of practical learning dynamics for this theoretically sound variant of TD-JEPA for future work.

## D  ADDITIONAL RESULTS

This section reports additional experiments, as well as detailed numerical results for plots in the main part of the paper.

### D.1  EXPLICIT STATE ENCODERS FOR ZERO-SHOT BASELINES

As mentioned in Section 6, existing zero-shot methods do not necessarily learn explicit state embeddings Wu et al. (2019); Touati & Ollivier (2021); Park et al. (2024). Nevertheless, we find that introducing a state encoder tends to improve average zero-shot performance, even in the absence of a specific representation learning objective. We consider three established zero-shot baselines (Laplacian (Wu et al., 2019), FB (Touati & Ollivier, 2021) and HILP (Park et al., 2025a)), and compare their performance without an explicit state encoder (i.e., successor features are trained on raw states), to those attained when introducing a state encoder (either shallow or deep). Note that, in this case, the state encoder is trained through gradient flowing from the original loss, and is not coupled to an ad-hoc objective. Results are summarized in Tab. 2, which only considers proprioceptive domains as a state encoder is necessary when learning from pixels. We observe that, while having an explicit state encoder may be detrimental in few specific domains, it remains beneficial on average, and crucial to obtain better performance in some domains. The optimal depth of the encoder is however domain-specific: OGBench domains generally prefer a deeper encoder, while DMC domains can be solved with a shallow encoder, or no explicit encoder at all. In order to maximize performance of the baselines, we thus evaluate them coupled with a deep encoder in OGBench, and with a shallow one in DMC (e.g., in Tab. 1). We additionally summarize performance differences induced by this choice of encoders in Fig. 5 (*left*).

### D.2  CONTRASTIVE VARIANT OF SYMMETRIC TD-JEPA

In order to further isolate the effect of different objectives on zero-shot performance, we instantiate the symmetric variant of TD-JEPA described by Eq. 7, where a single representation is used both as state and task encoder, as well as its contrastive counterpart, which uses an objective similar to

| | FB | FB$_{\text{shallow}}$ | FB$_{\text{deep}}$ | HILP | HILP$_{\text{shallow}}$ | HILP$_{\text{deep}}$ | Laplacian | Laplacian$_{\text{shallow}}$ | Laplacian$_{\text{deep}}$ |
|---|---|---|---|---|---|---|---|---|---|
| DMC (avg) | **648.9** ± 7.5 | **648.2** ± 4.1 | 632.0 ± 4.4 | **659.4** ± 4.1 | 620.1 ± 8.4 | 603.2 ± 5.8 | 585.6 ± 8.9 | 591.1 ± 10.7 | **598.8** ± 7.5 |
| walker | 788.7 ± 4.3 | **811.5** ± 5.9 | 812.4 ± 12.9 | 783.6 ± 8.4 | 796.4 ± 7.7 | 785.0 ± 9.1 | 754.6 ± 16.7 | 769.7 ± 4.7 | 762.1 ± 7.8 |
| cheetah | 662.8 ± 5.7 | 672.7 ± 4.9 | 635.7 ± 22.3 | 635.2 ± 9.4 | 618.3 ± 5.8 | 612.3 ± 12.2 | 640.5 ± 9.1 | 614.5 ± 18.9 | 630.7 ± 17.9 |
| quadruped | 574.1 ± 11.3 | 595.6 ± 9.1 | 590.6 ± 10.6 | 695.8 ± 13.5 | 694.8 ± 11.0 | 681.1 ± 16.2 | 590.6 ± 31.0 | 635.0 ± 38.7 | 651.5 ± 12.7 |
| pointmass | 570.0 ± 22.6 | 513.0 ± 20.0 | 489.3 ± 14.8 | 522.8 ± 18.9 | 371.0 ± 37.1 | 334.5 ± 27.9 | 356.7 ± 24.0 | 345.1 ± 22.4 | 351.0 ± 28.5 |
| OGBench (avg) | 19.07 ± 0.65 | 21.96 ± 0.81 | **39.04** ± 0.66 | 30.51 ± 1.20 | 35.02 ± 0.92 | **37.98** ± 1.11 | 9.07 ± 0.71 | 9.29 ± 0.59 | **14.81** ± 1.32 |
| antmaze-mn | 45.20 ± 2.05 | 49.00 ± 2.13 | **73.00** ± 2.72 | 62.20 ± 2.39 | 60.22 ± 2.57 | **83.60** ± 2.63 | 17.00 ± 2.72 | 25.80 ± 3.48 | 50.00 ± 4.94 |
| antmaze-ln | 19.80 ± 2.62 | 15.60 ± 2.54 | **36.80** ± 4.28 | 34.00 ± 2.86 | 27.40 ± 4.00 | **52.60** ± 3.86 | 13.60 ± 2.68 | 10.60 ± 2.46 | 21.60 ± 3.90 |
| antmaze-ms | 20.60 ± 5.38 | 32.40 ± 3.18 | **70.40** ± 3.95 | 12.00 ± 3.88 | 43.60 ± 5.12 | **50.60** ± 2.46 | 14.20 ± 3.24 | 12.60 ± 3.31 | 21.40 ± 4.32 |
| antmaze-ls | 8.00 ± 2.65 | 15.80 ± 1.05 | **49.80** ± 5.64 | 4.20 ± 2.05 | 11.20 ± 1.04 | 12.20 ± 1.75 | 4.44 ± 2.30 | 4.00 ± 1.58 | **11.80** ± 1.47 |
| antmaze-me | 23.00 ± 3.20 | 25.80 ± 1.65 | **51.60** ± 2.65 | 6.67 ± 2.16 | 8.60 ± 1.33 | 2.00 ± 0.84 | 0.40 ± 0.40 | 1.00 ± 0.68 | 0.80 ± 0.61 |
| cube-single | 21.00 ± 2.52 | 26.80 ± 2.62 | **49.60** ± 3.83 | 84.00 ± 2.76 | **88.00** ± 2.49 | 74.20 ± 3.53 | 19.20 ± 2.11 | 18.20 ± 2.14 | 15.11 ± 1.49 |
| cube-double | **4.20** ± 0.96 | **4.00** ± 0.79 | 2.60 ± 0.43 | 27.11 ± 3.02 | **29.20** ± 2.83 | 20.00 ± 2.72 | 2.80 ± 0.53 | **3.40** ± 0.85 | 2.00 ± 0.42 |
| scene | 23.80 ± 2.28 | 22.00 ± 2.29 | 12.80 ± 1.61 | 42.20 ± 1.75 | **45.00** ± 2.88 | 43.80 ± 1.90 | 7.80 ± 1.31 | 5.80 ± 0.96 | 7.80 ± 1.28 |
| puzzle-3x3 | **6.00** ± 0.84 | **6.22** ± 0.97 | 4.80 ± 0.68 | 2.20 ± 0.55 | 2.00 ± 0.52 | **2.80** ± 0.68 | 2.20 ± 0.63 | 2.20 ± 0.63 | **2.80** ± 0.68 |

Table 2: Ablation over encoder depth on proprioceptive DMC and OGBench domains for zero-shot baselines. Each method is evaluated in three variants: a baseline without explicit state encoder, one with a *shallow* (i.e., linear) state encoder, one with a *deep* one (2 or 4 hidden layers for DMC and OGBench, respectively, see Tab. 5.) We report mean performance and standard error. Results for each variant are bold if their confidence intervals overlap with the best variant of the same method.

---

**Algorithm 2** Symmetric TD-JEPA for zero-shot RL (latent-predictive and contrastive variants)

---

**Inputs**: Dataset $\mathcal{D}$, batch size $B$, regularization coefficient $\lambda$, networks $\pi, T_\phi, \phi$
Initialize target networks: $T_\phi^- \leftarrow T_\phi, \phi^- \leftarrow \phi$
**while** not converged **do**

   ▷ Sample training batch
   $\{(s_i, a_i, s_i')\}_{i=1}^B \sim \mathcal{D}, \{z_i\}_{i=1}^B \sim \mathcal{Z}, \{a_i'\}_{i=1}^B \sim \overline{\{\pi(\phi^-(s_i'), z_i)\}}_{i=1}^B$

   ▷ Compute latent-predictive/contrastive loss
   $\widehat{\mathcal{L}}(\phi, T_\phi) = \frac{1}{2B} \sum_i \left\| T_\phi(\phi(s_i), a_i, z_i) - \overline{\phi^-(s_i')} - \gamma \overline{T_\phi^-(\phi^-(s_i'), a_i', z_i)} \right\|^2$
   $\widehat{\mathcal{L}}(\phi, T_\phi) = \frac{1}{2B(B-1)} \sum_{i \neq j} \left( T_\phi(\phi(s_i), a_i, z_i)^\top \phi(s_j') - \gamma \overline{T_\phi(\phi(s_i'), a_i', z_i)^\top \phi(s_j')} \right)^2$
       $- \frac{1}{B} \sum_i T_\phi(\phi(s_i), a_i, z_i)^\top \phi(s_i')$

   ▷ Compute orthonormality regularization loss
   $\widehat{\mathcal{L}}_{\text{REG}}(\phi) = \frac{1}{2B(B-1)} \sum_{i \neq j} (\phi(s_i)^\top \phi(s_j))^2 - \frac{1}{B} \sum_i \phi(s_i)^\top \phi(s_i)$

   ▷ Compute actor loss
   $\{\hat{a}_i\}_{i=1}^B \sim \{\pi(\phi(s_i), z_i)\}_{i=1}^B$
   $\widehat{\mathcal{L}}_{\text{actor}}(\pi) = -\frac{1}{B} \sum_{i=1}^B T_\phi(\phi(s_i), \hat{a}_i, z_i)^\top z_i$

   Update $\phi, T_\phi$ to minimize $\widehat{\mathcal{L}}(\phi, T^\phi) + \lambda \widehat{\mathcal{L}}_{\text{REG}}(\phi)$
   Update $\pi$ to minimize $\widehat{\mathcal{L}}_{\text{actor}}(\pi)$
   Update target networks $\phi^-, T_\phi^-$ via EMA of $\phi, T_\phi$

---

that in Touati & Ollivier (2021). These algorithms are described in Alg. 2, where blue and yellow lines are exclusive to the latent-predictive and contrastive variants, respectively. Intuitively, the contrastive variant can be seen as a specific, symmetric instantiation of FB (Touati & Ollivier, 2021), and the comparison to the symmetric variant parallels that between TD-JEPA and FB. We report performance differences between the symmetric variant of TD-JEPA and the symmetric-contrastive variant in Fig. 5 (*right*). Similarly to results from Fig. 2, we observe that gaps in performance between self-predictive and contrastive methods grow larger when learning directly from pixels. For a numerical comparison, see Tab. 3.

## D.3 ADDITIONAL FAST ADAPTATION RESULTS

We extend the empirical evaluation on fast adaptation in Section 6 through Fig. 6. In the top part, we repeat the evaluation of Fig. 4, while only initializing encoders to pre-trained weights. The remaining components of actor and critic need to be learned from scratch. These experiments thus recall the standard setting in which pre-trained visual representations are evaluated (Nair et al., 2022; Ma et al., 2023; Majumdar et al., 2023). We observe that, while initial performance is near-zero for all methods, pre-trained representations maintain their effectiveness in terms of sample efficiency. In the middle row, we again repeat the evaluation from Fig. 4, but only freeze convolutional weights

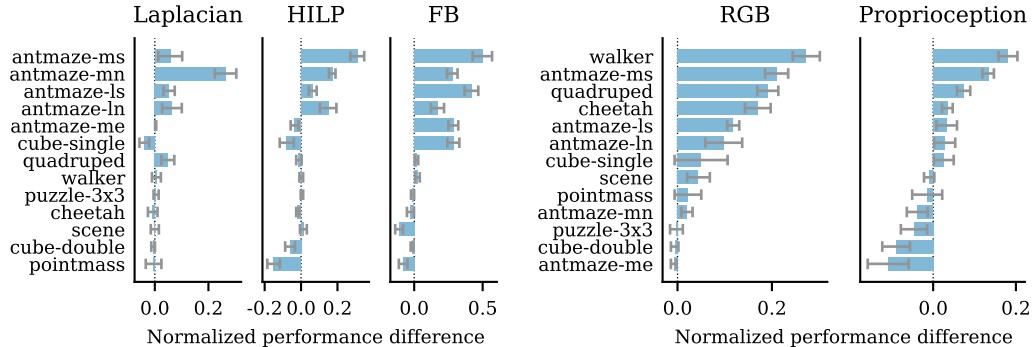

Figure 5: Difference in normalized performance between zero-shot baselines with and without an explicit encoder (**left**); normalized performance difference between symmetric TD-JEPA and its contrastive variant (**right**). Error bars represent standard errors on normalized performance differences.

for the variants represented by a dashed line. We observe that this causes the performance gap to full fine-tuning to shrink significantly, suggesting that convolutional filters extract suitable representations for both TD-JEPA and FB.

Finally, we extend our empirical evaluation to OGBench in the bottom part of Figure 6, in which we present results for the most challenging tasks in three representative domains. This evaluation differs from the previous ones, as we found it to require strong BC regularization, even during fine-tuning. We confirm that pre-trained representations remain beneficial in terms of sample efficient adaptation. Interestingly, we find that frozen representations pre-trained may outperform full fine-tuning in `antmaze-ls`, while they remained a bottleneck in DMC.

### D.4 ARCHITECTURAL ABLATIONS

Architectural choices are often crucial for self-predictive learning, and capacity is usually carefully distributed between encoder and predictor (Guo et al., 2022). This section ablates the width of the three main components in TD-JEPA in a controlled setting, namely in two visual OGBench tasks (`antmaze-ln` and `antmaze-ls`), which were chosen due to their complexity, and the fact that they often reward different approaches (see Tab. 1). We measure zero-shot performance as widths change in Fig. 7. In general, we observe that the state encoder (on the $y$ axis) should be as deep as possible (although this trend is not present in DMC, see Table 2). Instead, the task encoder (on the $x$ axis) should only be as deep as needed: at least 1 layer in `antmaze-ln` and closer to 4 for `antmaze-ls`. Finally, we observe that the predictor may be shallow, as long as the encoders have sufficient capacity. This result matches the general conjecture that latent-predictive representations should capture the key aspects of the input, to the extent that a prediction problem, e.g. predicting successor features, may be solved with limited capacity.

### D.5 VISUALIZATION OF TD-JEPA REPRESENTATIONS

Given a state-action pair $(s, a)$ and a further state $g$, one may easily evaluate the successor measure in $g$ of the policy that tries to reach $g$ from $s$: for instance, in the case of TD-JEPA, $M^{\pi_{\phi(g)}}(g \mid s, a) \approx T_\phi(\phi(s), a, z_g)^\top \psi(g)$, where $z_g = \mathbb{E}_{s \sim \mathcal{D}_{\text{rwd}}}[\psi(s)\psi(s)^\mathsf{T}]^{-1}\psi(g)$. Intuitively, this is connected to how quickly the policy may reach the goal, and should reflect the dynamics of the MDP. As often done in the literature (Lawson et al., 2025), we consider the methods evaluated in Tab. 1 and visualize these estimates for visual `antmaze-ln` in Fig. 8, highlighting how representations reflect temporal distances in the environment. While all plots reveal a similar pattern (which is not surprising as successor features are temporally-consistent representations by definition), some methods show inconsistent temporal distances. For instance, BYOL, RLDP, and HILP have latents of few states far from the goal with higher similarity to the latter than closer states. This is a sign of poor performance, as the predictor would be optimistic in predicting the number of steps to reach those goals.

A slightly different visualization of similarities is also helpful in highlighting the differences in the representation learning objectives of TD-JEPA, BYOL$^\star$ and BYOL-$\gamma^\star$. Figure D.4 displays similarities between the task embedding of a given goal $\psi(g)$, again depicted as a star, and predictions

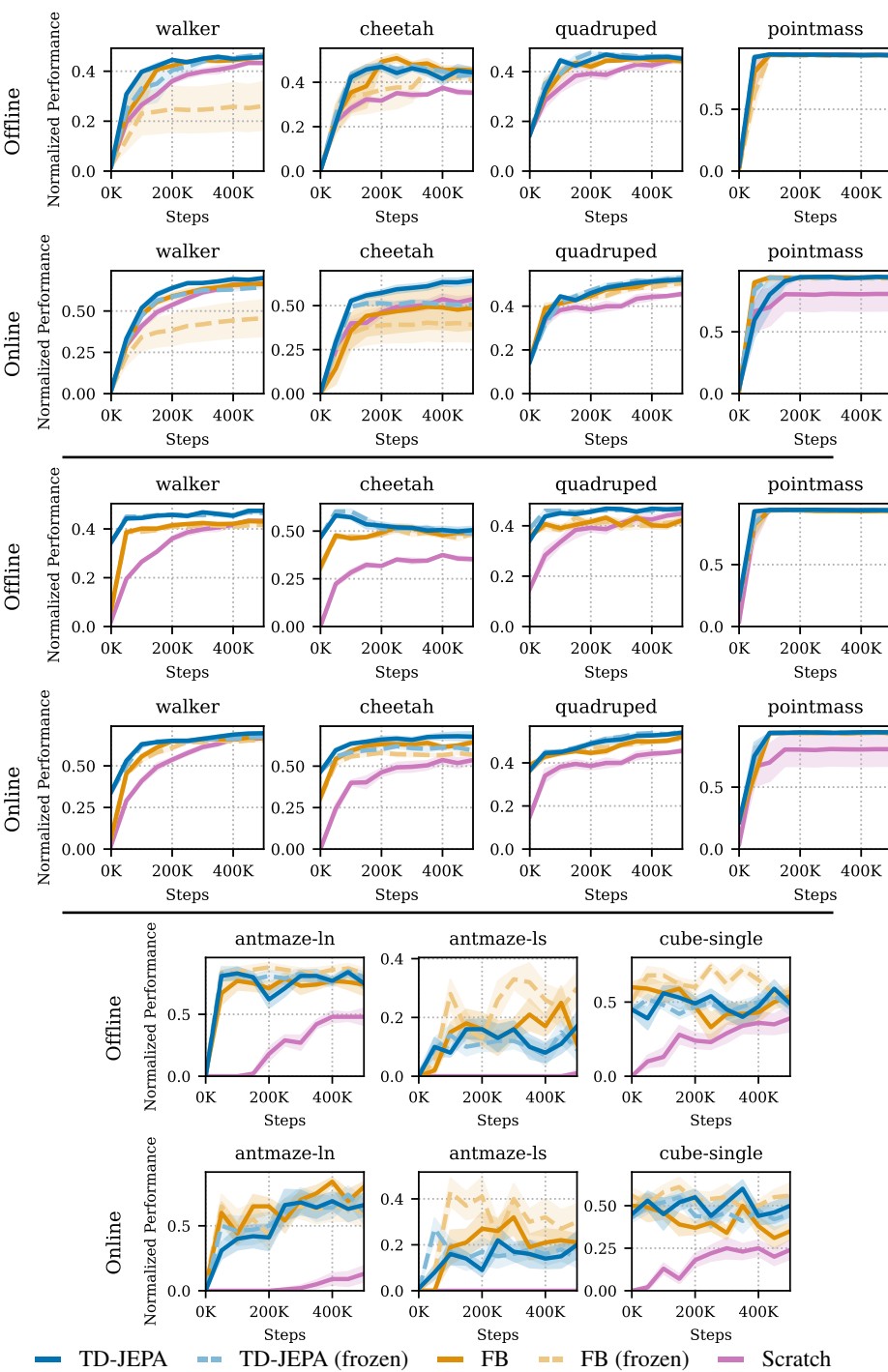

Figure 6: Additional fast adaptation results, obtained when only loading encoder's weights (**top**), when only freezing convolutional layers (**middle**) and in OGBench (**bottom**).

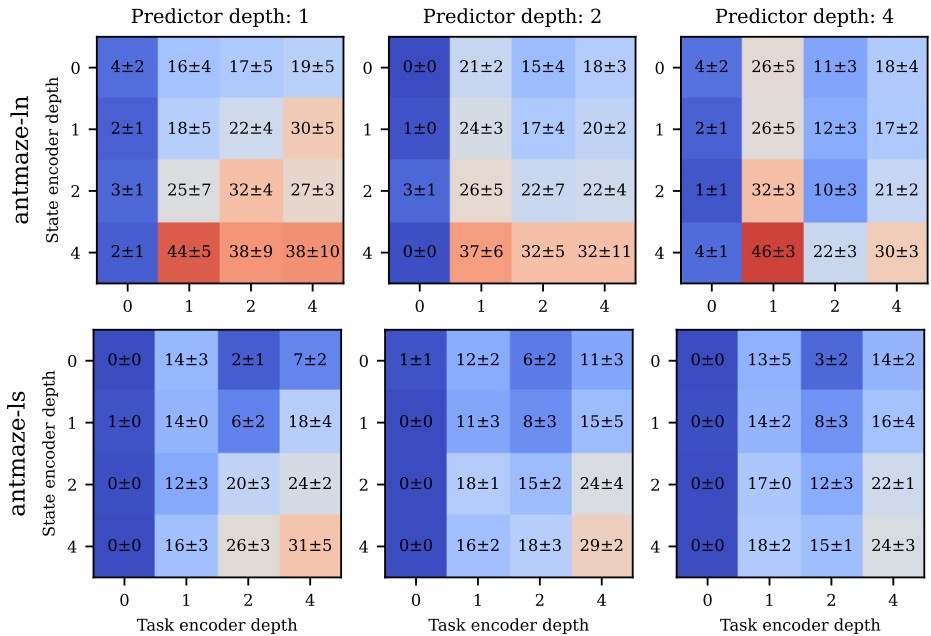

Figure 7: Zero-shot performance of TD-JEPA in `antmaze-ln` (top) and `antmaze-ls` (bottom) as the number of hidden layers in the encoders and predictors varies (from $0$ to $4$ and from $1$ to $4$, respectively).

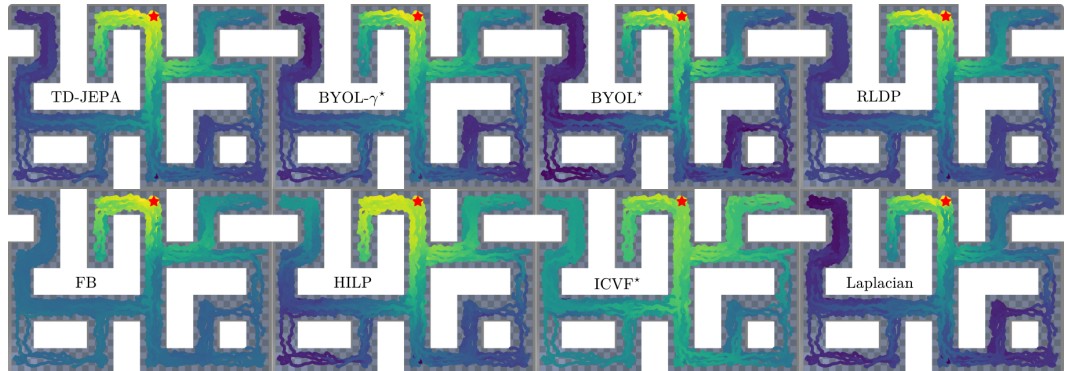

Figure 8: Cosine similarities between successor features and features (e.g., $\langle T_\phi(\phi(\cdot), a, z_g), \psi(g) \rangle$ for TD-JEPA) projected over $xy$ position of the agent's center of mass. The red star marks $g$. Similarities reflect shortest-path distances in the MDP.

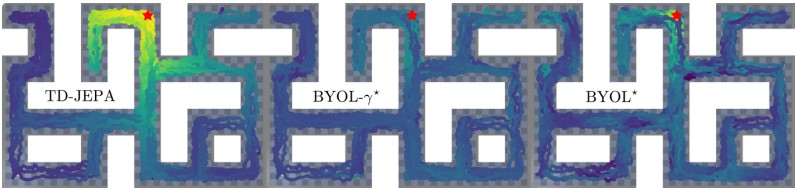

Figure 9: Cosine similarities between predictions and features (i.e., $\langle T_\phi(\phi(\cdot), a, z_g), \psi(g) \rangle$ for TD-JEPA and $\langle T(\phi(\cdot), a), \psi(g) \rangle$ for BYOL-based methods) projected over the $xy$ position of the agent's center of mass. The red star marks $g$. Similarities reflect alignment between predictions from a given state and the goal $g$. BYOL-based approaches model dynamics induced by the (unconditional) behavioral policy, and their predictions are not directed towards specific tasks.

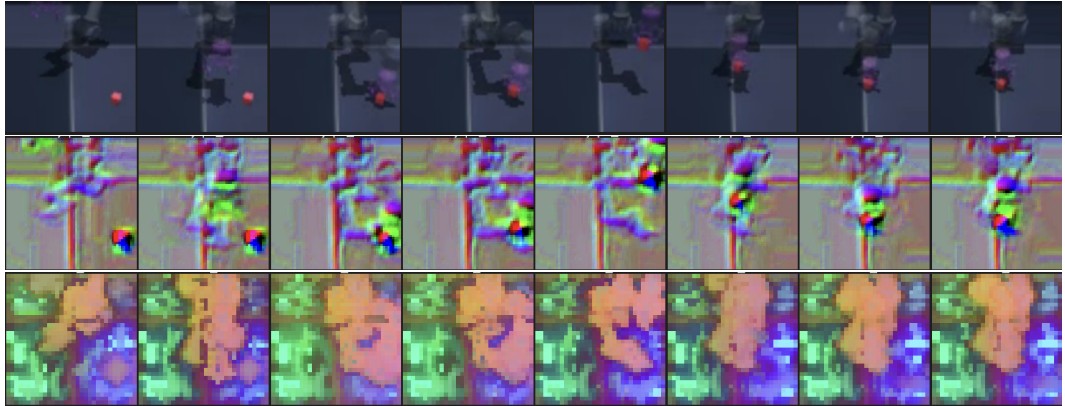

Figure 10: *Top:* a trajectory from the visual cube-single dataset of OGBench. *Middle:* visual features learned by TD-JEPA, computed as a PCA projection of the channels returned by the convolutional layers in $\phi$ down to 3 dimensions. *Bottom:* the same PCA projection applied to the visual features computed by DINO-v2.

from several state-action pairs in the dataset. In the case of TD-JEPA, this prediction is conditioned on the goal reaching task $z_g = \mathbb{E}_{s \sim \mathcal{D}_{\text{rwd}}}[\psi(s)\psi(s)^{\mathsf{T}}]^{-1}\psi(g)$, and we thus again visualize $T_\phi(\phi(s), a, z_g)^{\top}\psi(g)$. BYOL-based methods extract their representations through *unconditional* prediction of the very next state (BYOL$^{\star}$) or of a future state (BYOL-$\gamma^{\star}$). For these two algorithms, we thus may only compute predictions that are not conditioned on any particular task, and visualize their similarities $T(\phi(s), a)^{\top}\psi(g)$. As expected, the predictions of BYOL$^{\star}$ align well with the goal $g$ only near $g$ itself due to the one-step nature of the loss, while those of BYOL-$\gamma^{\star}$ are more "averaged out" since the algorithm models the long-term occupancy of a very stochastic behavior policy. Only TD-JEPA yield predictions that model the directed behavior of a task-aware policy.

We finally compare the visual features learned by TD-JEPA with those of DINO-v2 (Oquab et al., 2023), chosen as a representative method for general purpose (i.e., not control/dynamics-aware) vision encoders. As shown in Figure D.5 for the OGBench `cube-single` task, while DINO-v2 nicely segments the robotic arm (i.e., the main object in the scene), TD-JEPA focuses more on the end-effector and the cube (i.e., the objects that are most descriptive of the agent's behaviors). This showcases one of the advantages of (long-term) dynamics prediction, as focusing on those features that are most relevant for control.

### D.6 NUMERICAL RESULTS FOR PERFORMANCE DIFFERENCE PLOTS

We supplement the performance difference plot in Fig. 3 (right) and Fig. 5 (right) with Tab. 3, detailing numerical results.

### D.7 COMPUTATIONAL COST

We report the training speed (in terms of gradient steps per second) of all tested algorithms in Table 4. As expected, the symmetric variant of TD-JEPA (see Section 3.1) is generally the fastest algorithm, since it trains a single encoder and predictor through a simple TD loss. The asymmetric version of TD-JEPA is slower, requiring two encoders and two predictors, but is notably as fast, or faster than successor-feature-based methods like BYOL and BYOL-$\gamma$.

### D.8 PERFORMANCE ON LOW-QUALITY LOW-COVERAGE DATA

The zero-shot RL experiments of Section 6 focus on two types of datasets: low-quality high-coverage (ExoRL), where trajectories cover the state-action space well, but do not contain any "purposeful" behavior, and high-quality low-coverage (OGBench), where trajectories are mostly expert-like, but cover a narrow portion of the state space. Here we analyze TD-JEPA's performance under a more extreme setting of low-quality low-coverage data. To do so, we follow Jeen

| | C-TD-JEPA$_{sym}$ | TD-JEPA$_{sym}$ | TD-JEPA |
|---|---|---|---|
| DMC$_{RGB}$ (avg) | 437.2 $\pm$ 9.8 | 598.1 $\pm$ 5.9 | **628.8** $\pm$ **5.5** |
| walker | 413.2 $\pm$ 16.1 | 685.9 $\pm$ 14.8 | **738.9** $\pm$ **3.5** |
| cheetah | 517.9 $\pm$ 31.3 | 688.0 $\pm$ 7.2 | **706.0** $\pm$ **4.1** |
| quadruped | 428.3 $\pm$ 21.6 | **606.7** $\pm$ **20.1** | **626.7** $\pm$ **13.6** |
| pointmass | 389.5 $\pm$ 17.2 | 411.6 $\pm$ 13.9 | **443.7** $\pm$ **10.9** |
| DMC (avg) | 586.3 $\pm$ 14.6 | **657.5** $\pm$ **3.6** | **661.2** $\pm$ **6.3** |
| walker | 757.8 $\pm$ 12.4 | **800.7** $\pm$ **4.7** | 785.2 $\pm$ 6.7 |
| cheetah | 583.3 $\pm$ 23.3 | 618.1 $\pm$ 11.3 | **688.7** $\pm$ **6.7** |
| quadruped | 565.5 $\pm$ 14.4 | **731.7** $\pm$ **17.3** | 691.4 $\pm$ 5.0 |
| pointmass | 438.8 $\pm$ 24.5 | **479.6** $\pm$ **11.1** | **479.3** $\pm$ **23.6** |
| OGBench$_{RGB}$ (avg) | 33.93 $\pm$ 0.67 | 39.74 $\pm$ 0.64 | **41.34** $\pm$ **0.45** |
| antmaze-mn | 93.80 $\pm$ 1.05 | **95.80** $\pm$ **1.09** | **96.67** $\pm$ **1.11** |
| antmaze-ln | 68.00 $\pm$ 3.46 | **77.80** $\pm$ **4.22** | **74.60** $\pm$ **3.35** |
| antmaze-ms | 61.20 $\pm$ 3.00 | **82.20** $\pm$ **1.80** | **84.40** $\pm$ **3.85** |
| antmaze-ls | 8.80 $\pm$ 0.90 | 20.60 $\pm$ 1.03 | **28.80** $\pm$ **2.50** |
| antmaze-me | **1.40** $\pm$ **0.52** | **0.60** $\pm$ **0.31** | 0.20 $\pm$ 0.20 |
| cube-single | **64.00** $\pm$ **5.70** | **69.00** $\pm$ **2.62** | **67.80** $\pm$ **3.67** |
| cube-double | **2.00** $\pm$ **0.67** | 1.40 $\pm$ 0.60 | **3.00** $\pm$ **0.91** |
| scene | 4.00 $\pm$ 0.79 | 8.00 $\pm$ 1.94 | **14.20** $\pm$ **2.22** |
| puzzle-3x3 | **2.20** $\pm$ **0.76** | **2.22** $\pm$ **0.78** | **2.40** $\pm$ **0.83** |
| OGBench (avg) | 35.58 $\pm$ 0.97 | 35.20 $\pm$ 0.49 | **37.98** $\pm$ **0.77** |
| antmaze-mn | **82.20** $\pm$ **2.87** | 76.40 $\pm$ 2.65 | 70.40 $\pm$ 3.72 |
| antmaze-ln | **57.20** $\pm$ **4.47** | 43.60 $\pm$ 2.44 | **57.20** $\pm$ **4.25** |
| antmaze-ms | **74.00** $\pm$ **2.27** | 62.80 $\pm$ 3.59 | 61.56 $\pm$ 4.53 |
| antmaze-ls | **43.20** $\pm$ **3.16** | **41.40** $\pm$ **3.82** | **40.60** $\pm$ **2.51** |
| antmaze-me | **21.80** $\pm$ **3.30** | 17.00 $\pm$ 2.65 | **20.20** $\pm$ **2.39** |
| cube-single | 16.80 $\pm$ 2.15 | 20.00 $\pm$ 2.17 | **34.20** $\pm$ **2.88** |
| cube-double | **3.60** $\pm$ **1.07** | **2.40** $\pm$ **0.78** | **3.60** $\pm$ **0.78** |
| scene | 16.80 $\pm$ 1.31 | **39.40** $\pm$ **2.17** | 38.44 $\pm$ 1.37 |
| puzzle-3x3 | 4.60 $\pm$ 1.19 | **13.80** $\pm$ **1.47** | **15.60** $\pm$ **1.11** |

Table 3: Performance of TD-JEPA and symmetric variants (contrastive and latent-predictive) in DMC (returns) and OGBench (success rate) with either proprioception or RGB inputs. We report means and standard errors across seeds. Numbers are bold for top algorithms if confidence intervals overlap.

| | DMC$_{RGB}$ | DMC | OGBench$_{RGB}$ | OGBench |
|---|---|---|---|---|
| TD-JEPA | 19 | 52 | 35 | 132 |
| TD-JEPA (sym) | 34 | 84 | 61 | 161 |
| FB | 23 | 72 | 45 | 146 |
| BYOL | 19 | 55 | 36 | 112 |
| BYOL-$\gamma$ | 18 | 55 | 34 | 110 |
| HILP | 16 | 62 | 29 | 111 |
| Laplacian | 16 | 59 | 31 | 112 |
| ICVF | 11 | 55 | 26 | 99 |
| RLDP | 15 | 55 | 32 | 137 |

Table 4: Average number of iterations (gradient steps) per second of each algorithm for the experiments in Table 1. All methods are trained on the same hardware (a single V100 GPU) and use similarly sized architectures (see Appendix E). One training iteration samples a batch from the dataset and updates all networks once.

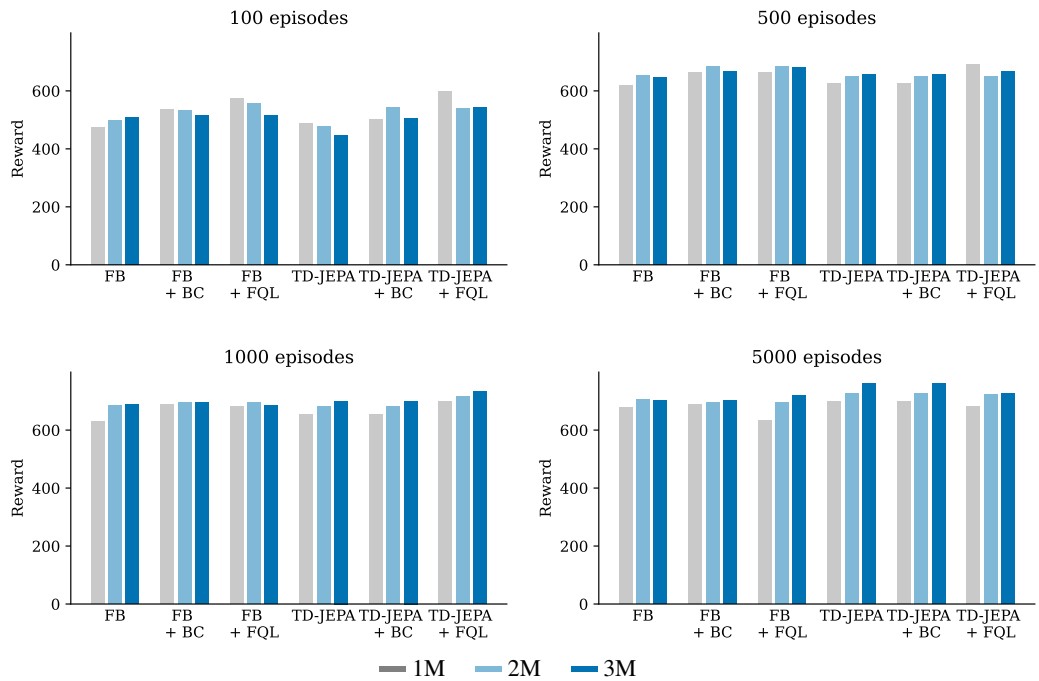

Figure 11: Average performance over all DMC `walker` and `quadruped` tasks for varying dataset sizes and number of training steps (1M, 2M, or 3M).

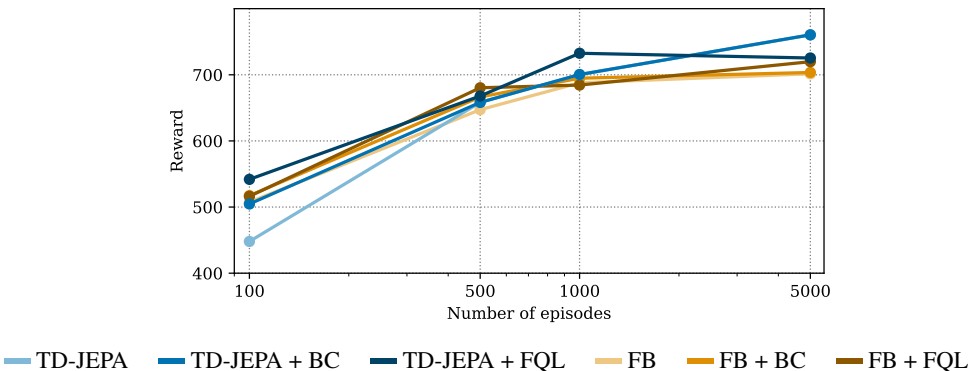

Figure 12: Average performance over all DMC `walker` and `quadruped` tasks as a function of the dataset size. All algorithms are trained for 3M steps.

et al. (2024a) and test TD-JEPA on two representative domains from ExoRL (walker and quadruped) while decreasing the dataset size. It is well known that offline training on low-coverage data requires some form of constraints to prevent extrapolation beyond the data distribution. We thus further report the performance of TD-JEPA combined with plain behavior-cloning (BC Fujimoto & Gu, 2021) and FQL-style (FQL Park et al., 2025b) regularization, the latter of which is used across OGBench experiments. The results, which also include FB-based variants, are shown in Figure D.8 and D.8. We can make the following observations: (1) as expected, the performance of all algorithms degrade when decreasing the data size, with drops consistent to those observed by Jeen et al. (2024a). (2) Both BC and FQL regularization are effective in mitigating the drops. For instance, the performance of TD-JEPA on the smallest dataset increases between 10% and 20% when adding regularization. (3) On small datasets, performance tend to decrease as we increase the number of training steps, an expected sign of overfitting. While behavior regularization does not mitigate this decrease, other techniques (e.g., early stopping) are likely to be beneficial. Finally, we remark that TD-JEPA could

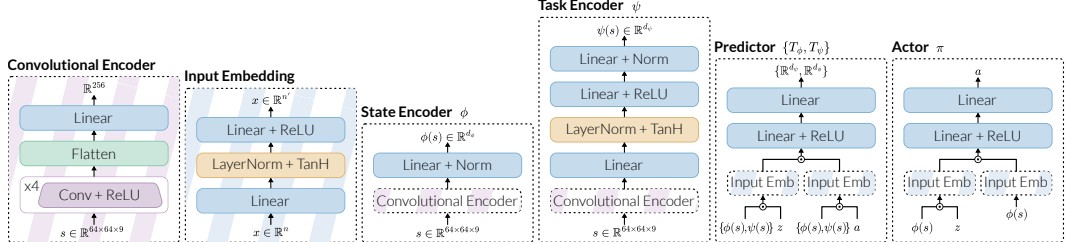

Figure 13: Overview of the architectures used by TD-JEPA in DMC$_{\text{RGB}}$. We refer to Tab. 5 for the different instantiations in other domains.

be easily combined with other conservative methods, such as advantage weighting or CQL (Jeen et al., 2024a). Evaluating these techniques goes beyond the scope of our work and represents and interesting avenue for future investigations.

# E  IMPLEMENTATION DETAILS

We organize the discussion of implementation details in several subsections.

## E.1  ENVIRONMENTS

Zero-shot results in Section 6 consider the standard Deepmind Control Suite (DMC, Tassa et al. (2018)) evaluation over four domains from (Touati et al., 2023): `walker`, `cheetah`, `quadruped` and `maze`. The first three define several locomotion tasks (e.g. `walk`, `run`, `flip`), while the latter evaluates both goal- and reward-based tasks. We additionally extend the zero-shot evaluation to the more recent OGBench suite (Park et al., 2024), which focuses exclusively on goal reaching. For computational reasons, we consider nine representative domains. We thus evaluate navigation across five `antmaze` datasets (`medium-{navigate, stitch, explore}` and `large-{navigate, stitch}`) [10], and manipulation through `cube-{single, double}`, `scene` and `puzzle-3x3`. In each domain we consider the five default tasks; for consistency with DMC, we carry out zero-shot evaluation through the standard reward inference procedure, and thus define each task through its reward function. During inference, we shift each reward by $+1$, which we found to significantly improve zero-shot performance.

## E.2  LEARNING FROM PIXELS

In visual experiments, environment return states as $64 \times 64$ RGB images. In order to alleviate non-Markovianity, states are composed by stacking 3 frames. Each image is scaled to $[-0.5, 0.5]$ and undergoes random shifts with a maximum intensity of 2 pixels, which is slightly milder than common strategies (Yarats et al., 2022b). The images are passed through convolutional encoders before being processed by further networks. Namely, we instantiate two convolutional encoders: one for the state encoder, and one for the task encoder (except for the symmetric variant of TD-JEPA, which only uses one). We however found that using a single, shared convolutional encoder does not significantly impact performance, and do not exclude that further specialization of the encoder might improve it. Each convolutional encoder uses the DrQ-v2 architecture introduced in Yarats et al. (2022b); we briefly experimented with IMPALA- (Espeholt et al., 2018) and Dreamer-like (Hafner et al., 2020) architectures, and we found them to achieve similar results when well-tuned.

## E.3  ARCHITECTURES

All algorithms rely on three types of networks, each of which is instantiated to the standard architectures from (Touati et al., 2023): (i) successor feature estimators, predictors and $F$-networks are MLPs with two layer-normalized embedding layers , (ii) state/task encoders and $B$-networks are standard MLPs with L2-normalized output and (iii) actor networks are Gaussian MLPs with a

---

[10]For compactness, we use `mn`, `ms`, `me`, `ln`, `ln` as dataset abbreviations, respectively.

| | DMC$_{\text{RGB}}$ | DMC | OGBench$_{\text{RGB}}$ | OGBench |
|---|---|---|---|---|
| SFs: $T_\phi, T_\psi, F$ - hidden layers | 3 | 3 | 4 | 4 |
| SFs: $T_\phi, T_\psi, F$ - hidden width | 1024 | 1024 | 512 | 512 |
| State Encoder: $\phi$ - hidden layers | 0 | 0 | 0 | 4 |
| State Encoder: $\phi$ - hidden width | 256 | 256 | 512 | 512 |
| State Encoder: $\phi$ - output dimension $d_\phi$ | 256 | 256 | 256 | 256 |
| Task Encoder: $\psi$ - hidden layers | 2 | 2 | 4 | 4 |
| Task Encoder: $\psi$ - hidden witdh | 256 | 256 | 512 | 512 |
| Task Encoder: $\psi$ - output dimension $d_\psi$ | 50 | 50 | 50 | 50 |
| Actor: $\pi$ - hidden layers | 3 | 3 | 4 | 4 |
| Actor: $\pi$ - hidden witdh | 256 | 256 | 512 | 512 |
| Discount factor $\gamma$ | 0.98 | 0.98 | 0.99 | 0.99 |
| Total gradient steps | 2M | 2M | 1M | 1M |
| Batch size $B$ | 512 | 1024 | 256 | 256 |
| Default learning rate | $10^{-4}$ | $10^{-4}$ | $10^{-4}$ | $10^{-4}$ |
| Default regularization coefficient $\lambda$ | 1.0 | 1.0 | 1.0 | 1.0 |
| EMA coefficient | 0.001 | 0.001 | 0.005 | 0.005 |
| $p_{\text{goal}}$ | 0.5 | 0.5 | 0.5 | 0.5 |

Table 5: Architectural (top) and training (bottom) hyperparameters.

similar architecture to predictors, and fixed standard deviation of 0.2. All networks use ReLU activations, except for embedding layers, which use TanH. The number and width of layers in DMC closely follow those described in Touati et al. (2023), while we utilize deeper, narrower networks in OGBench, in order to better align with the implementation of benchmarked methods in Park et al. (2025a). An overview of the architectures used for TD-JEPA in DMC$_{\text{RGB}}$ is depicted in Fig. 13; further architectural hyperparameters describing depth and width of these networks are reported in the first half of Table 5. We remark that the state encoder's depth in proprioception was found to be quite impactful, and was tuned according to baseline performance, as shown in Table 2.

### E.4 TRAINING HYPERPARAMETERS

The second part of Table 5 describes hyperparameters used for training, and is complemented by the following discussion of further details. We again closely follow the default hyperparameters from Touati et al. (2023) and Park et al. (2025a) whenever possible; the batch size is reduced in visual domains to meet computational limitations. All networks are optimized through Adam (Kingma, 2014). TD-targets and value estimates for SVG-like policy updates are computed as the *mean* of twin networks; latents $z \in \mathcal{Z}$ are representations of random uniform states from the dataset $\psi(s)$ with probability $p_{\text{goal}}$, and are sampled from the hypersphere (i.e., $\mathcal{Z}$) otherwise (Touati et al., 2023).

### E.5 METHOD-SPECIFIC DETAILS

The baselines' implementation closely follows the public codebases, when available.

Our implementation of **FB** builds upon the one released in Tirinzoni et al. (2025), which is in turn aligned with the code released by Touati et al. (2023).

As described in Jajoo et al. (2025), **RLDP** is implemented in the same framework, but the latent dynamics model and the task encoder are trained in parallel with the remaining components; gradients from the contrastive FB objective are not backpropagated through the task encoder. As in the original work, we found that multi-step prediction results in better performance, and we similarly adopt a prediction horizon of $H = 5$. Given a batch of trajectories $\{(s_0^i, a_0^i, \ldots, s_H^i)\}_{i=0}^{B-1}$, a task encoder $\psi$ and latent predictor $T_{\text{RLDP}} : \mathbb{R}^{d_\psi} \times \mathcal{A} \to \mathbb{R}^{d_\psi}$, the loss for training task representations is thus:

$$\widehat{\mathcal{L}}_{\text{RLDP}}(T_{\text{RLDP}}, \psi) = \frac{1}{2B} \sum_{i=0}^{B-1} \sum_{t=0}^{H-1} \left\| h_t^i - \psi^-(s_t^i) \right\|^2, \tag{28}$$

where $h_0^i = \psi(s_0^i)$, $h_t^i = T_{\text{RLDP}}(h_{t-1}^i, a_{t-1}^i)$, and $\psi^-$ is a target network. The latent predictor is instantiated with the hyperparameters presented for the SFs architecture described in Tab. 5.

**Laplacian** (Mahadevan & Maggioni, 2007), **HILP** (Park et al., 2024), **BYOL$^\star$** (Grill et al., 2020), **BYOL-$\gamma^\star$** (Lawson et al., 2025) and **ICVF$^\star$** (Ghosh et al., 2023) are all implemented in a successor-feature framework: the losses proposed in the original publications are optimized to retrieve a task encoder $\psi$. Considering a batch of transitions $\{(s_i, a_i, s_i')\}_{i=0}^{B-1}$, the feature learning objectives for Laplacian, BYOL$^\star$ and BYOL-$\gamma^\star$ are, respectively:

$$\widehat{\mathcal{L}}_{\text{Laplacian}}(\psi) = \frac{1}{2B} \sum_{i=0}^{B-1} \|\psi(s_i) - \psi(s_i')\|^2, \tag{29}$$

$$\widehat{\mathcal{L}}_{\text{BYOL}}(T_{\text{BYOL}}, \psi) = \frac{1}{2B} \sum_{i=0}^{B-1} \left\|T_{\text{BYOL}}(\psi(s_i), a) - \psi^-(s_i')\right\|^2, \tag{30}$$

$$\widehat{\mathcal{L}}_{\text{BYOL}-\gamma}(T_{\text{BYOL}}, \psi) = \frac{1}{2B} \sum_{i=0}^{B-1} \left\|T_{\text{BYOL}}(\psi(s_i), a) - \psi^-(s_i^+)\right\|^2, \tag{31}$$

$$\tag{32}$$

where $T_{\text{BYOL}} : \mathbb{R}^{d_\psi} \times \mathcal{A} \to \mathbb{R}^{d_\psi}$ is a jointly trained latent predictor, $s_i^+ \sim M^{\pi_\beta}(\cdot|s_i, a_i)$, $\pi_\beta$ is the behavioral policy, and the minus sign $(-)$ denotes target networks. HILP and ICVF instead train representations through

$$\widehat{\mathcal{L}}_{\text{HILP}}(\psi) = \frac{1}{2B} \sum_{i=0}^{B-1} \ell_\tau^2(-\mathbf{1}_{s_i \neq g_i} - \gamma\|\psi_-(s') - \psi_-(g)\| + \|\psi(s) - \psi(g)\|), \tag{33}$$

$$\widehat{\mathcal{L}}_{\text{ICVF}}(T_{\text{ICVF}}, \phi, \psi) = \frac{1}{2B} \sum_{i=0}^{B-1} |\tau - \mathbf{1}_{A_i < 0}| \big(V(s_i, g_i, y_i) - \mathbf{1}_{s_i = g_i} - \gamma V_-(s_i', g_i, y_i)\big)^2, \tag{34}$$

where $\ell_\tau^2(x) = |\tau - \mathbf{1}_{x<0}|x^2$ is an expectile regression loss with expectile $\tau$, $g_i$ and $y_i$ are goals (or intents) sampled from a mixture of future and random states as described in Ghosh et al. (2023), $V(x, y, z) = \phi(x)^\top T_{\text{ICVF}}(\psi(z))\psi(y)$, $T_{\text{ICVF}} : \mathbb{R}^{d_\psi} \to \mathbb{R}^{d_\psi \times d_\psi}$ is a matrix predictor and $A_i = \mathbf{1}_{s_i = y_i} + \gamma V(s_i', y_i, y_i) - V(s_i, y_i, y_i)$. In an unified way across baselines, universal successor feature estimators $F_\psi : \mathbb{R}^{d_\phi} \times \mathcal{A} \times \mathcal{Z} \to \mathbb{R}^{d_\psi}$ (and state encoders $\phi : \mathcal{S} \to \mathbb{R}^{d_\phi}$) are then trained through standard TD-learning over features $\psi$ by optimizing:

$$\widehat{\mathcal{L}}_{\text{SF}}(F_\psi, \phi) = \frac{1}{2B} \sum_{i=0}^{B-1} \left\|F_\psi(\phi(s_i), a_i; z_i) - \overline{\psi^-(s_i')} - \gamma\overline{F_{\psi,-}(\phi^-(s_i'), a_i'; z_i)}\right\|^2, \tag{35}$$

where $a_i' \sim \pi(\phi(s_i'), z_i)$. The latent dynamics model in BYOL$^\star$ and BYOL-$\gamma^\star$, as well as the intent-conditioned predictor from ICVF$^\star$ (Ghosh et al., 2023), share the architecture of SFs, as described in Table 5, potentially dropping the subnetworks that receive states, actions or latents $z$ as necessary. The state encoder in ICVF$^\star$ receives gradients from both the ICVF loss, and the successor feature prediction, which we found to slightly improve performance compared to only propagating gradients in either direction.

We found all zero-shot algorithms to be sensitive to certain hyperparameters, such as the strength of orthonormal regularization $\lambda$. For a fair comparison, we evaluate all algorithms over a small hyperparameter grid (6 configurations in DMC, and 4 in OGBench), and report performances for the best performing configuration for each domain. For all algorithms, we sweep over two values for the learning rate of the task encoder $\psi$: $[10^{-4}, 10^{-5}]$. A second important hyperparameter is the orthonormal regularization coefficient $\lambda$, for which optimal ranges strongly differ across algorithms. In order to avoid an excessively large sweep, the ranges were thus tuned for each algorithm on a representative subset of domains, and are reported in Table 6. In general, we observe that contrastive methods prefer very strong regularization, while self-predictive methods can learn with weaker regularization in certain domains. For algorithms that do not leverage orthonormal regularization (HILP, ICVF$^\star$), we instead sweep over the likelihood of sampling random goals/intents in $[0.375, 0.5]$.

|  | Laplacian | FB | RLDP | BYOL* | BYOL-$\gamma$* | TD-JEPA |
|---|---|---|---|---|---|---|
| DMC$_{\text{RGB}}$ | [0.01, 0.1, 1] | [0.01, 0.1, 1] | [0.01, 0.1, 1] | [0.01, 0.1, 1] | [0.01, 0.1, 1] | [0.01, 0.1, 1] |
| DMC | [0.01, 0.1, 1] | [0.01, 0.1, 1] | [0.01, 0.1, 1] | [0.01, 0.1, 1] | [0.01, 0.1, 1] | [0.01, 0.1, 1] |
| OGBench$_{\text{RGB}}$ (nav) | [0, 1] | [100, 1000] | [100, 1000] | [0.001, 0.01] | [0.001, 0.01] | [0.1, 1] |
| OGBench (nav) | [0, 1] | [100, 1000] | [100, 1000] | [0.001, 0.01] | [0.001, 0.01] | [0.1, 1] |
| OGBench$_{\text{RGB}}$ (man) | [0, 1] | [100, 1000] | [100, 1000] | [0.01, 0.1] | [0.01, 0.1] | [1, 10] |
| OGBench (man) | [0, 1] | [100, 1000] | [100, 1000] | [0.01, 0.1] | [0.01, 0.1] | [1, 10] |

Table 6: Orthonormal regularization ranges for each algorithm. (nav) and (man) indicate navigation and manipulation domains, respectively.

### E.6 OFFLINE CORRECTION

The standard zero-shot evaluation pipeline (Touati et al., 2023) has traditionally relied on high-coverage datasets (Yarats et al., 2022a). OGBench (Park et al., 2025a) represents an interesting challenge in this sense, as most datasets are expert-like, and the incomplete support over actions induces well-known offline issues (Kumar et al., 2020). While these problems have been previously studied in the context of zero-shot RL (Jeen et al., 2024b), we find that a novel instantiation of regularization techniques for single-task RL works well in this setting. In particular, we rely on a FlowQ-like regularization scheme (Park et al., 2025b): we train a flow model to estimate the behavioral policy, and replace the Gaussian policy with a noise-conditioned, one-step policy. This policy is regularized to samples generated by the flow model through 10 integration steps. The resulting behavior cloning loss term is normalized by the mean absolute Q-value in the batch, and scaled by a regularization coefficient $\alpha$: we use $\alpha = 3$ and $\alpha = 0.3$ for manipulation and navigation tasks, respectively. Both networks are instantiated with the architectural hyperparameters described for $\pi$ in Table 5. Finally, in order to be able to track BC targets, the policy is trained directly over the state space (or on the output of convolutional encoders in visual domain), instead of acting over state representations produced by state encoders $\phi(\cdot)$.

### E.7 EVALUATION: ZERO-SHOT AND FAST ADAPTATION

While zero-shot evaluation is averaged over all tasks (4-7, depending on the domain), fast adaptation is only evaluated on the hardest task per domain, estimated through average zero-shot performance of TD-JEPA and FB. This corresponds to `run` in `walker`, `cheetah`, `quadruped` and `square` in `maze`; we evaluate task 2 in `antmaze` domains and task 4 in `cube-single`. Fast adaptation methods that rely on pre-trained weights are initialized from the best performing zero-shot run. Behavior cloning regularization is carried over from pre-training if present, but orthonormal regularization is no longer applied. For online adaptation, the Update-to-Data ratio is fixed to 1, and $5000$ initial transitions are collected by the policy before fine-tuning. Batches are sampled from a 50-50 mixture of the relabeled pre-training dataset, and a replay buffer containing the most recent $2 \cdot 10^5$ transitions.

The procedure to extract an actor and critic that are trainable through TD3 (Fujimoto et al., 2018) from zero-shot agents is rather direct. First, the relabeled dataset is used to infer the optimal latent $z_r$ through linear regression. Then, the critic network may then be derived from successor feature estimates: in the case of FB (Touati & Ollivier, 2021), $Q(s, a) \approx F(\phi(s), a, \overline{z_r})^\top \overline{z_r}$; in the case of TD-JEPA the same holds as soon as $F$ is replaced by $T_\phi$. Although adaptation schemes over $\mathcal{Z}$ are possible (Sikchi et al., 2025), $z_r$ can be kept frozen. In order to match the output of the critic to the scale of rewards, $z_r$ is scaled such that its squared L2 norm matches the maximum reward over the dataset. Finally, the actor $\pi(s)$ can be directly extracted from zero-shot policies as soon as they are conditioned on $\overline{z_r}$.

All evaluation metrics are averaged over 10 episodes in OGBench, and 20 in DMC, and computed over 10 random seeds for OGBench and 5 for DMC.

### E.8 PSEUDOCODE FOR TD-JEPA

```python
def train(self):
    # sample training batch
    obs, action, next_obs = self.replay_buffer.sample()
    z = self.sample_z(obs)

    # compute targets
    next_phi = self.target_phi_encoder(next_obs)
    next_psi = self.target_psi_encoder(next_obs)
    next_action = self.actor(next_phi).sample()
    target_phi = next_psi + discount * self.target_phi_predictor(next_phi, z, next_action)
    target_psi = next_phi + discount * self.target_psi_predictor(next_psi, z, next_action)
    # compute predictions
    phi = self.phi_encoder(obs)
    psi = self.psi_encoder(obs)
    pred_phi = self.phi_predictor(phi, z, action)
    pred_psi = self.psi_predictor(psi, z, action)
    jepa_loss = mse(pred_phi - target_phi.detach()) + mse(pred_psi - target_psi.detach())
    # regularize
    phi_cov = torch.matmul(phi, phi.T)
    phi_ortho_loss = - phi_cov.diag().mean() + 0.5 * phi_cov.off_diag().pow(2).mean()
    psi_cov = torch.matmul(psi, psi.T)
    psi_ortho_loss = - psi_cov.diag().mean() + 0.5 * psi_cov.off_diag().pow(2).mean()

    # compute actor loss
    actor_action = self.actor(phi.detach(), z).sample()
    actor_pred = self.phi_predictor(phi.detach(), z, actor_action)
    actor_loss = (actor_pred * z).sum(-1).mean()

    # aggregate losses and optimize
    loss = jepa_loss + self.lambda_phi * phi_ortho_loss + self.lambda_psi * psi_ortho_loss
    loss += actor_loss
    self.optimizer.zero_grad()
    loss.backward()
    self.optimizer.step()
    # update target networks
    update_target(self.target_phi_encoder, self.phi_encoder)
    update_target(self.target_psi_encoder, self.psi_encoder)
    update_target(self.target_phi_predictor, self.phi_predictor)
    update_target(self.target_psi_predictor, self.psi_predictor)
```

Listing 1: Python-like pseudocode for TD-JEPA (simplified).

Pseudocode for the default variant of TD-JEPA is shown in Listing 1; the output of predictors is assumed to be averaged across twin networks.

