# OpenReview forum: "TD-JEPA: Latent-predictive Representations for Zero-Shot Reinforcement Learning"
_ICLR.cc/2026/Conference — ICLR 2026 Oral_

### Official Review · Reviewer_ghUc · 2025-10-24

**Soundness:** 4
**Presentation:** 4
**Contribution:** 3
**Rating:** 8
**Confidence:** 4

**Summary:**

This paper proposes a new state representation learning method called TD-JEPA. The main idea of TD-JEPA is to use TD learning to learn BYOL/JEPA-like self-predictive representations, where positives are sampled from geometrically distributed future states (in the MC case). Importantly, instead of learning representations w.r.t. the behavioral policy (like BYOL-$\gamma$), the authors simultaneously learn a latent task embedding $z$ and train a $z$-conditioned policy and the corresponding representations, somewhat similarly to FB representations. The authors show that TD-JEPA representations enable zero-shot RL, outperforming previous zero-shot RL approaches on ExORL and OGBench. They also demonstrate that these representations can be effectively fine-tuned for offline-to-online RL.

**Strengths:**

This is a well-written paper with solid theoretical and empirical results. The proposed method is (to my knowledge) novel, even though there are a number of closely related (but different) works, such as FB and BYOL-$\gamma$. The authors compare their method with previous methods across diverse categories, and convincingly demonstrate its effectiveness on a wide array of tasks and settings.

Another strength is that the authors provide a solid theoretical analysis of their method. They theoretically show that their method (more or less) low-rank approximates successor measures. With this connection, they are able to relate downstream performance to their loss. While I didn't exhaustively check the correctness of the proofs in Appendix, they appear to be solid and have some new, intriguing aspects of their own.

**Weaknesses:**

I don't see any major weaknesses in this work. While I believe the current results are already enough for an ICLR publication, I think they could be further strengthened by demonstrating the performance on even more complex, "new" environments that go beyond the standard benchmarks used in previous work (Motivo is a good example of this, and I think this could also be done in a separate follow-up work). Another nitpick is that the related work section is placed in Appendix. Especially given that this area is (relatively) dense, I believe discussions about related work are essential to understanding this method, and would like to encourage the authors to move this section to the main paper in the final version.

**Questions:**

- In Table 4, why do some of the settings have zero hidden layers?
- While I understood that the TD-JEPA representation approximates the successor measure of $\pi^z$ in a low-rank manner ($F_z^\top B$) for a **fixed** $z$, can the authors explain what kind of behavior **set** this method would learn? In other words, are there any explicit descriptions of learned skills? For example, for HILP, one can describe its skills as those that maximally span the isometric latent embedding space. Do the authors have a similar intuitive (yet "correct") description of skills learned by TD-JEPA? (I guess this question directly applies to FB representations as well.)

---

> ### Author Response · Authors · 2025-11-23
>
> We would like to thank the reviewer for their thorough assessment of our submission, and for the interesting comments, which we address one by one.
>
> >  I think they could be further strengthened by demonstrating the performance on even more complex, "new" environments [...]
>
> We agree that more complex environments would represent a promising next step in this direction. While TD-JEPA tackles a fundamental representation learning problem, it can be seamlessly integrated in the Motivo framework, retaining the policy regularization component and inheriting the architectural improvements (see also the answer to Reviewer WTAu). While we are excited to follow this direction, we believe that experiments at such a larger scale deserve significant efforts, and an independent investigation that we leave for future work.
>
> > [...] the related work section is placed in Appendix
>
> We are happy to use the majority of the additional page granted for revisions to host related works, which are now featured as Section 5.
>
> > In Table 4, why do some of the settings have zero hidden layers?
>
> We use zero hidden layers to denote a linear map. Note, however, that the only setting where $\phi$ is actually just a matrix multiplication (followed by L2-normalization) is proprioceptive DMC, in which we found that all baselines generally preferred shallower state encoders. In this setting, TD-JEPA with a deeper encoder performed similarly well, and a shallower one was used for simplicity and uniformity. In the case of proprioceptive OGBench, we instead found that all methods preferred deeper state encoders.
> In pixel-based experiments, the state encoder is actually a DrQ-style deep convolutional network, and Table 4 only reports the architecture of the MLP head added on top to project features to a lower dimension (i.e., a linear projection is sufficient for this purpose).
> In summary, the optimal depth of the state encoders was domain- (but not algorithm-) dependent, and Table 1 reports results for the best performing variants.
>
> > [...] can the authors explain what kind of behavior set this method would learn?
>
> This is a great question, which we believe poses several technical challenges. For a fixed set of features, TD-JEPA extracts the same behaviors that successor-feature methods (such as FB or HILP) would: it learns policies that are optimal for all rewards that lie in the feature’s span. However, there is a feedback loop between the feature learning objectives and policy optimization: features are trained to decompose policy-conditional occupancy measures. Similarly to FB, we know that if this decomposition is perfect, TD-JEPA learns optimal policies for any possible reward function (see Theorem 4 and comments below). However, just like for FB, a more precise characterization remains challenging when this decomposition is approximate, and deserves further study.
>
> ---
>
> Thank you again for your detailed evaluation and actionable suggestions. We hope we were able to address every comment; we would be happy to engage in any further discussion.

---

> > ### Comment · Reviewer_ghUc · 2025-11-24
> >
> > Thanks for the response. I have no additional concerns and would like to happily maintain my original score.

---

### Official Review · Reviewer_FzbF · 2025-10-26

**Soundness:** 4
**Presentation:** 4
**Contribution:** 4
**Rating:** 8
**Confidence:** 3

**Summary:**

The authors propose TD-JEPA, a method for learning policy-dependent representations of long-term dynamics that can be used for zero-shot RL. They learn the representations by combining temporal difference methods with the self-predictive framework of JEPA. Across a number of experimental benchmarks, they show that TD-JEPA matches or outperforms existing baselines, especially on pixel-based environments. The authors also discuss a number of theoretical properties of the proposed algorithm.

**Strengths:**

The paper is well written and clearly motivates the proposed algorithm. The experiments are comprehensive and show convincing performance. The proofs justify the algorithm theoretically and build on techniques used in previous works.

**Weaknesses:**

I do not have any major weaknesses for this work, but I point out things I found interesting which could benefit from more detailed discussion (although I understand some of these points might be outside of the scope of this work).

1. What is the computational cost of the different proposed methods? I know many of the considered methods train multiple function approximators, but it would be nice to have at least a brief discussion of the training speed for each method.
2. I found the comparison between BYOL and TD-JEPA interesting. In particular, the contrast between modeling expert policies vs. policy conditional measures. I wonder if there are other simple analyses/visualizations that can pinpoint this down further, beyond performance on benchmarks. For example, is one representation more robust to noise / generalizes better (perhaps since TD-JEPA is better on pixel environments, this would be the case)?
3. For pixel-based environments, how do the visual features compare from using a purely visual pre-training strategy (MAE for example) to learning the encoders with TD-JEPA (or another RL method)?

In general, there seem to be a number of different "representation learning" methods that are useful for RL in the community. The authors already briefly do this in the discussion section, but continued conversation about the long-term desiderata for a representation, beyond performance on zero-shot benchmarks would be interesting for the community.

**Questions:**

Please see above.

---

> ### Author Response · Authors · 2025-11-23
>
> We would like to thank you for the thorough and supportive feedback. We were able to further expand on the suggested points in the revision.
>
> > What is the computational cost of the different proposed methods?
>
> We have added a thorough discussion of training speed (in terms of steps per second) as Appendix D.7 (Table 4). Due to its simplicity, the symmetric variant of TD-JEPA is the fastest among the methods we benchmarked; the asymmetric version is in line with existing zero-shot methods.
>
> > I found the comparison between BYOL and TD-JEPA interesting [...] I wonder if there are other simple analyses/visualizations.
>
> Analyzing the gap between TD-JEPA and BYOL methods beyond current results is not straightforward; earlier preliminary experiments on robustness to noise did not display significant differences, which we believe makes sense since there is no component in these algorithms that would make one more robust than another. The main difference is about what the representations are trained to capture. To better highlight this point, we added an additional visualization to Appendix D.5 in the revision. Contrary to Figure 8, which displayed similarities between successor features and task representations, this new visualization directly measures correlation between the output of the predictor and task representations, thus displaying more clearly the role of training objectives for TD-JEPA, BYOL-\gamma and BYOL (e.g., what it means to be task-conditioned and long-term).
>
> > [...] how do the visual features compare from using a purely visual pre-training strategy (MAE for example)?
>
> This is another interesting point, which allows comparing control-aware with control-unaware representations. Following this suggestion, we have added a visualization of the features extracted through DINOv2 – chosen as a representative of purely visual pre-training strategies – and TD-JEPA in Appendix D.5. This highlights how the features extracted by the latter better focus on image details relevant for control.
>
> > [...] continued conversation about the long-term desiderata for a representation, beyond performance on zero-shot benchmarks would be interesting for the community.
>
> Following your suggestion, we have tried to condense the following paragraph and incorporate it into the discussion (Section 7); we hope it will provide some interesting guidance for future works.
>
> *Our method tackles a fundamental representation learning problem for control, and highlights a connection between downstream performance and accurate modeling of the successor measure. We thus suggest that flexible representations for RL, particularly for value estimation and optimization on downstream tasks, should be predictive of future behaviors, and precisely capture their diverse and long-term nature.*
>
> ---
>
> We would like to thank the reviewer again, and hope that our revision was able to address each of the points raised. We remain available for any further discussion.

---

> > ### Comment · Reviewer_FzbF · 2025-11-24
> >
> > Thank you for answering my questions. I maintain that this is a good paper that should be accepted.

---

### Official Review · Reviewer_u6m8 · 2025-10-29

**Soundness:** 4
**Presentation:** 3
**Contribution:** 3
**Rating:** 6
**Confidence:** 3

**Summary:**

TD-JEPA targets learning latent policy dynamics models through successor features from a collection of reward-free off-policy data. Unlike prior work often limited to one-step predictions, this TD loss enables the model to learn representations predictive of long-term, policy-conditioned latent dynamics. The system trains four components directly in latent space: a state encoder ($\phi$), a task encoder ($\psi$), a policy-conditioned multi-step predictor ($T_\phi$), and a set of policies ($\pi_z$). The predictor learns to approximate successor features, which allows the agent to perform zero-shot optimization of any new reward function at test time.

**Strengths:**

TD-JEPA introduces a method for learning long-term transition dynamics rather than single-step dynamics. There is extensive experimentation in simulated benchmarks across DMC and OGBench showing general improvement over other zero-shot RL baselines.

**Weaknesses:**

Perhaps it is because I am not in the immediate area, but it was challenging for me to determine what the real-world impact of this method is. It certainly makes sense in relation to recent latent prediction models, but the introduction may benefit from some framing that takes a step back. Is there more application-facing work that has called for learning from large reward-free multi-task offline datasets that could be cited? And for zero-shot RL? If this doesn’t exist, why not yet?

**Questions:**

In the abstract, it is written “(TD) learning enables learning representations predictive of long-term latent dynamics across multiple policies…” . Would it be more accurate to say “long-term latent task/policy dynamics” instead?

For me, the notational assignments were difficult to maintain as I was reading through the relatively terse section 3. I wonder if there is a more distinguishing notation between \psi and \phi?

---

> ### Author Response · Authors · 2025-11-23
>
> Thank you for your assessment of our submission, and for your suggestions. We will address each of the points individually.
>
> > Is there more application-facing work that has called for learning from large reward-free multi-task offline datasets that could be cited? And for zero-shot RL?
>
> This is an important point to raise, and we are happy to comment on it. Several real-world tasks can benefit from training on large, reward-free, multi-task offline datasets.
> Current efforts in robotics control, and in particular manipulation, have identified in-the-wild videos as a promising source of information to tap into. These data sources typically include demonstrations of several tasks, and are largely reward-free [1,2]. Recent works have pushed this direction significantly, either by leveraging action annotation [3,4] or latent actions [5]. Other works in this space focus on large teleoperation datasets, which are again multi-task and reward-free. This is particularly the case of modern VLAs (see e.g. [13]), which have achieved impressive applications in the real world.
>
> Important developments have taken place in whole-body humanoid control. In this area, agents are normally trained on motion-capture datasets, which can be regarded as multi-task unlabeled data, with recent works achieving impressive results on real hardware [10,11,12]. The main limitation of these methods is that they focus exclusively on imitating the behaviors in the data (e.g., via motion tracking). Unsupervised zero-shot RL methods have recently stepped in as a promising solution, enabling agents that can quickly solve tasks beyond imitation, like reward optimization and goal reaching. They have been successfully applied in both simulation [6] and on real hardware [7].
>
> These works suggest that there is a practical opportunity for unsupervised zero-shot RL; as suggested, we have added this discussion to the extended Related Works section in App. A and referred to these applications in the introduction.
>
> > Would it be more accurate to say “long-term latent task/policy dynamics” instead?
>
> We agree that clarifying the connection between policies and tasks makes this sentence clearer. We have reformulated it as “ [...] (TD) learning enables learning representations predictive of long-term latent dynamics across multiple policies/tasks“.
>
> > I wonder if there is a more distinguishing notation between \psi and \phi?
>
> We have resorted to using \psi and \phi to represent state/task representations in order to align with previous works [8, 9]; when experimenting with subscripts and superscripts, we found that they made the notation arguably more dense.
>
> ---
>
> We hope we were able to thoroughly describe related application-facing work, and to answer all other questions. Please, do not hesitate to let us know of any further comment.
>
> **References:**
>
> [1] Khazatsky et al., DROID: A Large-Scale In-The-Wild Robot Manipulation Dataset, RSS DGR 2024
>
> [2] Open X-Embodiment Collaboration. Open X-Embodiment: Robotic Learning Datasets and RT-X Models, ICRA 2024
>
> [3] Sobal et al., Learning from Reward-Free Offline Data: A Case for Planning with Latent Dynamics Models, arXiv 2025
>
> [4] Assran et al., V-JEPA 2: Self-supervised Video Models Enable Understanding, Prediction and Planning, arXiv 2025
>
> [5] Ye et al., Latent Action Pretraining from Videos, ICLR 2025
>
> [6] Tirinzoni et al., Zero-Shot Whole-Body Humanoid Control via Behavioral Foundation Models, ICLR 2025
>
> [7] Li et al., BFM-Zero: A Promptable Behavioral Foundation Model for Humanoid Control Using Unsupervised Reinforcement Learning, arXiv 2025
>
> [8] Ghosh et al., Reinforcement Learning from Passive Data via Latent Intentions, ICML 2023
>
> [9] Bhateja et al., "Robotic Offline RL from Internet Videos via Value-function Pre-training." arXiv 2023
>
> [10]  Zeng et al., Behavior Foundation Model for Humanoid Robots. arXiv 2025
>
> [11] He et al., Asap: Aligning Simulation and Real-world Physics for Learning Agile Humanoid Whole-body Skills. arXiv 2025
>
> [12] Weng et al., HDMI: Learning Interactive Humanoid Whole-Body Control from Human Videos. arXiv 2025.
>
> [13] GR Team. Gemini Robotics: Bringing AI into the Physical World. ArXiv 2025.

---

> ### Comment · Reviewer_u6m8 · 2025-11-27
>
> I thank the authors for the thorough response and have certainly learned more about the applications of the area. I do not feel confident enough to raise my score further. However, I maintain my positive opinion regarding the work.

---

### Official Review · Reviewer_WTAu · 2025-11-01

**Soundness:** 3
**Presentation:** 3
**Contribution:** 3
**Rating:** 8
**Confidence:** 3

**Summary:**

The paper proposes TD-JEPA, a latent-predictive representation learning method for zero-shot reinforcement learning (RL) that uses temporal difference (TD) learning to capture long-term multi-policy dynamics from offline, reward-free data. By training separate state/task encoders, policy-conditioned predictors, and parameterized policies in latent space, TD-JEPA enables zero-shot optimization of arbitrary reward functions—even for challenging pixel-based inputs. Theoretical guarantees and extensive experiments across 13 datasets (ExoRL/OGBench) validate its effectiveness.

**Strengths:**

1. Sound Motivation and Novel Method: The work addresses a key limitation of existing latent-predictive methods (single-task/one-step prediction, on-policy dependence) by leveraging TD learning for offline, multi-step, multi-policy dynamics modeling. Its design—separating state (low-level dynamics) and task (high-level context) encoders, plus TD-based losses and regularization—balances innovation and practicality, avoiding representation collapse while enabling offline training.
2. Comprehensive and Convincing Experiments: TD-JEPA is evaluated across 65 tasks (locomotion, navigation, manipulation) with proprioceptive/pixel inputs. It matches or outperforms SOTA baselines. Ablations confirm the value of multi-step prediction and separate encoders, while fast adaptation results show pre-trained representations boost sample efficiency for fine-tuning.
3. Rigorous Theoretical Foundations:
The paper provides solid theoretical support.

**Weaknesses:**

TD-JEPA relies on FlowQ-like behavioral cloning regularization to handle low-coverage datasets in OGBench, but its performance under more extreme data scarcity (e.g., critical action gaps, sparse trajectories) is not fully validated. It would help to add analysis on how TD-JEPA’s performance decays as data coverage decreases, and compare it to methods specifically designed for low-quality offline data. This would better demonstrate its practical applicability to real-world scenarios where data is often incomplete.

**Questions:**

In terms of methodology, could you further compare the approach proposed in this paper with that proposed by Motivo [1]?

[1] Andrea Tirinzoni, Ahmed Touati, Jesse Farebrother, Mateusz Guzek, Anssi Kanervisto, Yingchen Xu, Alessandro Lazaric, and Matteo Pirotta. Zero-shot whole-body humanoid control via behavioral foundation models. ICLR, 2025.

---

> ### Author Response · Authors · 2025-11-23
>
> We would like to thank the reviewer for their positive and thorough assessment of this work. We are happy to provide the additional empirical support and clarifications suggested.
>
> >  [...] performance under more extreme data scarcity (e.g., critical action gaps, sparse trajectories) is not fully validated
>
> FlowQ-like regularization is instrumental to performance on OGBench, which features expert-like data distributions: the quality of trajectories is usually high, but data support is limited. In order to validate performance under extreme data scarcity, we followed [1] (which studies exactly the setting of low-quality low-coverage data) and evaluated the zero-shot performance of TD-JEPA on increasingly smaller subsets of the ExoRL data (100, 500, 1000, and 5000 episodes). What we found is that the performance of TD-JEPA degrades consistently as what is observed in the literature (e.g., by [1] for FB), which is not surprising as this is a well-known issue of offline RL with low-coverage data. Interestingly, we also found that FlowQ-like regularization or even simple behavior-cloning regularization are effective in mitigating this drop, improving performance from 10% to 20% on the smallest data sizes. Finally, we note that TD-JEPA could be easily combined with other conservative methods used in offline RL, such as IQL, AWAC, or CQL. Evaluating them is an interesting direction for future work. We have added this study in Appendix D.8.
>
> > [...] could you further compare the approach proposed in this paper with that proposed by Motivo?
>
> Motivo studies a slighly different problem — how to improve zero-shot RL with the aid of action-free expert data — and it is directed towards a specific application (humanoid control). The solution proposed in Motivo is online training through a standard zero-shot RL algorithm (FB), with a discriminator-driven policy-regularization term that encourages the learned behaviors to match the state distribution of the expert data. On the other hand, our work investigates the fundamental question of how to learn good representations that enable zero-shot RL, and proposes a novel solution based on latent-predictive learning, with a particular emphasis on the importance of encoding states and tasks. In this sense, TD-JEPA can be directly compared to FB, since the two methods study the same problem (zero-shot RL). We have done so both conceptually and numerically (see, e.g., the Related Works section, which we moved to the main paper thanks to the increase in page limit). While online, regularized learning is beyond the scope of this work, a version of TD-JEPA which inherits the policy regularization mechanism from Motivo (essentially replacing FB as the main zero-shot component) is thus fully compatible with the online pipeline for learning humanoid control. Given the positive performance comparison we reported between TD-JEPA and FB in standard offline zero-shot benchmarks, this constitutes a very promising direction for future work.
>
> ---
>
> Thank you again for your review. We hope our additional evaluation under data scarcity addresses your concerns. We are happy to further discuss this point, or any other.
>
> **References:**
>
> [1] Jeen et al., Zero-shot Reinforcement Learning from Low-Quality Data, NeurIPS 2024

---

### Author Response · Authors · 2025-11-23
**General response**

We would like to thank all reviewers for the positive evaluation of our work, and for providing detailed and actionable feedback, which was instrumental in further improving our submission. We were happy to directly incorporate suggestions in our new revision; we have highlighted new parts in pink for convenience. Beside several clarifications, we have (i) added new visual comparisons to latent-predictive and control-unaware representation learning methods (D.5), (ii) provided a comparison of training speed (D.7), (iii) integrated an empirical ablation on low-quality, low-coverage data (D.8), and (iv) moved a detailed discussion of related works from the appendix to the main paper.

We hope we were able to clarify each and every comment, and we are happy to further discuss any questions that may remain.

---

### Meta-Review · Area_Chair_WosV · 2026-01-07

**Summary:**

The reviewers expressed strong consensus for accepting the paper, highlighting the novelty of integrating Temporal Difference (TD) learning with Joint Embedding Predictive Architectures (JEPA) for zero-shot reinforcement learning. Key strengths identified include the rigorous theoretical analysis of latent-predictive representations and the extensive empirical validation across 13 datasets (ExoRL, OGBench), where the method matches or outperforms state-of-the-art baselines. Concerns regarding performance on low-quality data, computational overhead, and comparisons to related workA were effectively addressed during the rebuttal, reinforcing the decision to accept.

**Reviewer Concerns:**

Reviewer Concerns addressed by the rebuttal:

Performance on low-quality data (Reviewer WTAu): The authors conducted additional experiments demonstrating that regularization mitigates performance degradation on small data subsets, alleviating concerns about robustness in data-scarce regimes.

Comparison to related work (Reviewer WTAu, ghUc): The distinction between TD-JEPA and Motivo was clarified and related work was moved to the main text.

**Reviewer Scores:**

All 4 reviewers will maintain their positive scores.

---

### Decision · Program_Chairs · 2026-01-26

Accept (Oral)